# Efficient Quantization of Mixture-of-Experts with Theoretical Generalization Guarantees

**Mohammed Nowaz Rabbani Chowdhury**[1]**, Kaoutar El Maghraoui**[2]**, Hsinyu Tsai**[2]**,
Naigang Wang**[2]**, Geoffrey W. Burr**[2]**, Liu Liu**[1]**, Meng Wang**[1,*]
[1]Rensselaer Polytechnic Institute, [2]IBM Research

## Abstract

Sparse Mixture-of-Experts (MoE) allows scaling of language and vision models efficiently by activating only a small subset of experts per input. While this reduces computation, the large number of parameters still incurs substantial memory overhead during inference. Post-training quantization has been explored to address this issue. Because uniform quantization suffers from significant accuracy loss at low bit-widths, mixed-precision methods have been recently explored; however, they often require substantial computation for bit-width allocation and overlook the varying sensitivity of model performance to the quantization of different experts. We propose a theoretically grounded expert-wise mixed-precision strategy that assigns bit-width to each expert primarily based on their *change in router's $l_2$ norm* during training. Experts with smaller changes are shown to capture less frequent but critical features, and model performance is more sensitive to the quantization of these experts, thus requiring higher precision. Furthermore, to avoid allocating experts to lower precision that inject high quantization noise, experts with large *maximum intra-neuron variance* are also allocated higher precision. Experiments on large-scale MoE models, including Switch Transformer and Mixtral, show that our method achieves higher accuracy than existing approaches, while also reducing inference cost and incurring only negligible overhead for bit-width assignment.[1]

## 1 Introduction

The sparse Mixture of Experts (MoE) architecture allows the construction of larger pre-trained language and vision models without increasing training costs (Shazeer et al., 2017; Lepikhin et al., 2021; Riquelme et al., 2021; Fedus et al., 2022; Allingham et al., 2022). In this architecture, the transformer block's feed-forward network (FFN) is replaced by multiple FFN modules, each referred to as an *expert*. Each expert is paired with a trainable *router*, which selectively activates a small subset of experts for each input token. Compared to dense models (Fedus et al., 2022; Chowdhury et al., 2023), MoE enables faster convergence and reduces the amount of training data required. Additionally, MoE maintains similar inference FLOPs to dense models despite having a larger parameter count (Riquelme et al., 2021; Zhou et al., 2022).

Despite these advantages, MoE incurs substantial memory costs during inference due to their large size, limiting their deployment. Since experts learn diverse features during pre-training, not all are equally relevant for a specific downstream task, some recent pruning strategies attempt to mitigate memory usage by eliminating task-irrelevant experts (Chen et al., 2022a; Koishekenov et al., 2023; Chowdhury et al., 2024). However, the effectiveness of pruning diminishes in complex tasks, where a larger set of experts remains essential.

Post-training weight quantization (PTWQ) focuses on quantizing weights after training and has emerged as another promising technique for reducing the memory footprint of large language models (LLMs) (Shao et al., 2024; Hubara et al., 2021; Lin et al., 2024; Frantar et al., 2023; Badri & Shaji, 2023). Several works have applied quantization to large MoE models by uniformly reducing all expert weights to a fixed bit-width (Kim et al., 2022; 2023b; Frantar & Alistarh, 2024). However, this uniform

---

*Corresponding author. Email: wangm7@rpi.edu.
[1]Code available at: `https://github.com/nowazrabbani/moe_quantization`

approach overlooks the varying importance of different experts, resulting in substantial performance degradation under extremely low-bit settings (e.g., sub-3-bit quantization). Although various mixed-precision quantization methods have recently been developed for other model architectures and could potentially be applied to MoEs, e.g., block-wise mixed precision of MoEs Li et al. (2024), these do not leverage the varying relevance of experts in MoE models. An *expert-wise* mixed-precision approach, in which bit-width varies across experts based on their sensitivity, offers greater potential to preserve accuracy under low-bit constraints. Yet, this direction remains largely unexplored. To our knowledge, only two recent works (Li et al., 2024; Huang et al., 2025a) have explored this approach, using metrics such as expert usage frequency and mean routing weights to estimate experts' sensitivity. However, these heuristics are suboptimal and lack theoretical justification (Chowdhury et al., 2024). This raises a fundamental question:

*What metric provably categorizes experts in the mixed-precision quantization of an MoE layer?*

This paper addressed the question both theoretically and empirically. Our theoretical analysis reveals that allocating higher bit-width to a group of experts with a **smaller** *change in the router's $l_2$ norm* during training, corresponding to experts that learn less frequently used but important features, while allocating the rest of the experts in lower bit-width can significantly reduce model size without hurting performance. Moreover, allocating some of the experts with high *maximum intra-neuron variance* to higher bit allows further compression. Extensive empirical evaluations support these findings, demonstrating that large state-of-the-art (SOTA) MoE models (e.g., Switch Transformer, Mixtral) can be quantized to ultra-low-bit regimes (e.g., below 3-bit) without sacrificing accuracy. Our major contributions are summarized as follows:

1. **A theoretically grounded metric, the change in a router's $l_2$ norm during training, for expert-wise mixed-precision quantization.** We theoretically analyze the training dynamics and generalization behavior of a simplified two-layer MoE model fine-tuned on classification tasks. This model is a SOTA theoretical model in understanding training and generalization of MoEs and general neural networks. We prove that experts capturing less prevalent features exhibit smaller changes in their router's $l_2$ norm during training. We further prove that these experts exhibit lower activation levels. Hence, the model's generalization performance is more sensitive to the quantization of these experts, requiring them to have higher precision. Unlike the prior work (Chowdhury et al., 2024) that uses the router's $l_2$ norm to distinguish between relevant and irrelevant experts for pruning, our analysis offers a finer-grained view that identifies varying levels of expert importance, enabling a principled approach to diverse expert-wise bit-width allocation for mixed-precision quantization.

2. **Empirical Validation of Expert-wise Mixed Precision on large MoE models, including Switch Transformer and Mixtral.** Our results show that assigning precision based on changes in the router's $l_2$ norm during fine-tuning outperforms alternative heuristics, such as expert activation frequency and activation weights. Moreover, for large pretrained models like Mixtral, where fine-tuning is computationally expensive, we demonstrate that using the router's $l_2$ norm from the pretrained model alone, without any fine-tuning, achieves test accuracy comparable to the existing expert-wise mixed-precision strategies (Table 1) while reducing inference computation (Figure 3). Importantly, our approach incurs negligible computational overhead to determine expert bit-widths, while the alternative methods require significant GPU computation.

## 2 RELATED WORKS

**Quantization of large models.** Parallel to PTWQ, some other methods focus on minimizing quantization error through quantization aware (re-)training (QAT), but these methods are every expensive and not suitable for large models (Wang et al., 2022; Liao et al., 2024; Gu et al., 2024). Mixed-precision strategies have also been explored, where bit-widths vary across different model components (e.g., MLP blocks, attention heads) (Dong et al., 2020; Li et al., 2024; Huang et al., 2025b; Dettmers et al., 2024). However, intra-layer variation of bit-widths remains underexplored.

**MoE compression.** To compress MoE models, some approaches focus on expert pruning, either targeting specific downstream tasks during fine-tuning (Chen et al., 2022a; Koishekenov et al., 2023; Chowdhury et al., 2024), or removing irrelevant experts from the pre-trained model (Zhang et al., 2024; Xie et al., 2024). However, the effectiveness of pruning diminishes for complex tasks. PTWQ has also been applied to compress large MoE models, with most methods focusing on uniform

quantization of experts (Kim et al., 2022; 2023a;b; Yi et al., 2025; Frantar & Alistarh, 2024), which often results in degraded performance under ultra low-bit settings. Two very recent works have explored expert-wise mixed-precision quantization for MoE models (Li et al., 2024; Huang et al., 2025a), but they either rely on suboptimal metrics or require extensive memory and computational resources to determine the expert-specific bit-width distribution.

**Optimization and generalization analysis of neural networks.** Several works have established optimization and generalization guarantees for neural networks (NNs) using neural tangent kernel (NTK)-based approaches (Jacot et al., 2018; Lee et al., 2019; Du et al., 2019; Allen-Zhu et al., 2019; Li et al., 2022). However, such analyses can not capture realistic training dynamics as they require the weights remain close to initialization throughout training. More recent studies have focused on the feature learning dynamics of NNs to derive generalization guarantees (Karp et al., 2021; Brutzkus & Globerson, 2021; Li et al., 2023; Zhang et al., 2023; Chowdhury et al., 2023), offering better alignment with practical neural network behavior. These analyses are typically restricted to shallow networks, and our work falls within this framework.

# 3 EXPERT-WISE MIXED-PRECISION QUANTIZATION OF MoE

## 3.1 THE BASICS OF MIXTURE-OF-EXPERTS ARCHITECTURE

In MoE models, the standard feed-forward networks (FFNs) in transformer MLP blocks are replaced with multiple parallel FFN *experts*. A gating network of *routers* assigns input tokens to specific experts.

Consider an example MoE block that includes $k$ experts, each of which is a two-layer FFN. Let $x = [x^{(1)^\top}, x^{(2)^\top}, ..., x^{(n)^\top}] \in \mathbb{R}^{nd}$ denote the input sequence, consisting of $n$ tokens, each of dimension $d$. For each token $x^{(j)} \in \mathbb{R}^d$ with $j \in [n]$, the MoE block produces a $d'$-dimensional output token, forming the output sequence $x_{out} = [x_{out}^{(1)^\top}, x_{out}^{(2)^\top}, ..., x_{out}^{(n)^\top}] \in \mathbb{R}^{nd'}$. The output $x_{out}^{(j)} \in \mathbb{R}^{d'}$ corresponding to the input $x^{(j)}$ is given by,

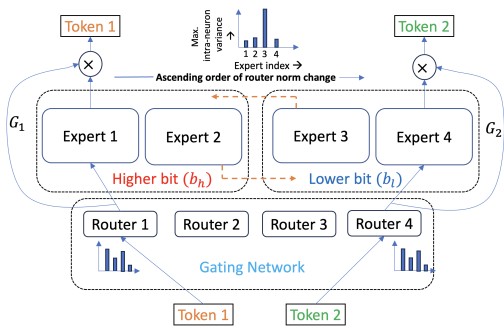

Figure 1: A schematic of Mixture-of-Experts with our proposed approach. Experts with smaller router norm changes in higher bit. Experts with large max intra-neuron variance are reordered.

$$x_{out}^{(j)} = \sum_{s \in [k]} f_s(x^{(j)}) \text{ where, } f_s(x^{(j)}) = \begin{cases} W_2^{(s)} \sigma\left(W_1^{(s)} x^{(j)}\right) G_j^{(s)} & \text{if } G_j^{(s)} > 0 \\ \vec{0}_{d'} & \text{if } G_j^{(s)} = 0 \end{cases} \quad (1)$$

Here $f_s(x^{(j)}) \in \mathbb{R}^{d'}$ denotes the contribution of the $s$-th expert. $W_1^{(s)} \in \mathbb{R}^{m \times d}$ and $W_2^{(s)} \in \mathbb{R}^{d' \times m}$ represent the weights of the first and second layer, respectively. The activation function $\sigma(\cdot)$ is applied element-wise to the hidden layer output. $\vec{0}_{d'}$ denotes the zero vector in $\mathbb{R}^{d'}$.

**The Gating Network.** For each token $x^{(j)}$ ($j \in [n]$) and each expert $s$ ($s \in [k]$), the gating network computes a *gating value* $G_j^{(s)} \in [0, 1]$. The network includes $k$ trainable router vectors $w_s \in \mathbb{R}^d$ ($s \in [k]$), one per expert.

In *token-choice routing*(Fedus et al., 2022), given an input token $x^{(j)} \in \mathbb{R}^d$, the routing network computes a set of *routing scores* $\{\langle w_s, x^{(j)} \rangle\}_{s=1}^k$ for all $k$ experts. The top-$l$ experts with the highest scores (where $l \ll k$) are selected, and their corresponding *gating values* are computed via a softmax over the top-$l$ scores, while the remaining experts receive a gating value of zero. In contrast, in *expert-choice routing* (Zhou et al., 2022), each expert $s$ computes routing scores $\{\langle w_s, x^{(j)} \rangle\}_{j=1}^n$ over all $n$ tokens and selects the top-$l$ tokens with the highest scores. The gating values for the selected tokens are computed via a softmax over the top-$l$ scores, and the rest are assigned zero.

## 3.2 The Post-Training Weight Quantization (PTWQ)

PTWQ methods compress neural network weights by representing them as low-bit fixed-point integers. During inference, these quantized weights are dequantized back to floating-point values. Given a weight matrix $W \in \mathbb{R}^{in \times out}$, and a target bit-width $b$, the de-quantized weights $\hat{W}$ are computed as,

$$\hat{W} = \Delta \cdot \left( \lfloor W/\Delta + z \cdot 1^{in \times out} \rceil - z \cdot 1^{in \times out} \right) \tag{2}$$

where $\Delta := (\max(W) - \min(W))/(2^b - 1)$ is the quantization bin size, and $z := -\lfloor \min(W)/\Delta \rceil - 2^{b-1}$ is the zero-point of the quantized weights. $1^{in \times out}$ is an all-ones matrix with dimension (in $\times$ out), and $\lfloor \cdot \rceil$ is the element-wise rounding to the nearest integer.

Most PTWQ methods select $\Delta$ and $z$ by minimizing either the loss in activation on calibration data (e.g., GPTQ (Frantar et al., 2023), AWQ (Lin et al., 2024)). An alternative line of work aims to minimize the reconstruction error directly, without calibration data (e.g., HQQ (Badri & Shaji, 2023)).

## 3.3 The Proposed Mixed-Precision Quantization Method

Our quantization approach proceeds in two main steps: (1) **ordering the experts** based on *the change in the router's norm* and *the maximum intra-neuron variance*, defined in Defs. 3.1 and 3.2 and (2) **assigning bit-widths** (two-level or three-level quantization) according to the ordering.

**STEP 1: Expert Ordering**. We first introduce two metrics used in ordering the experts.

**Definition 3.1.** *For expert $s \in [k]$, let $w_s^{(0)}$ and $w_s^{(T)}$ be its router vectors in the initial and trained models, respectively. We define* **change in router's** $l_2$ **norm** *as follows,*

$$\Lambda_s^{(T)} := \|w_s^{(T)}\| - \|w_s^{(0)}\|. \tag{3}$$

**Definition 3.2.** *Let $W_1^{(s,T)}$ be the first-layer weight matrix of expert $s$ in the trained model, containing $m$ neurons with weights in $\mathbb{R}^d$. The* **maximum intra-neuron variance** *evaluates the maximum variance of weight entries in each neuron, i.e.,*

$$\text{MaxVar}_s^{(T)} := \max_{r \in [m]} \frac{1}{d} \sum_{i=1}^{d} \left( W_1^{(s,T)}[r,i] - \tfrac{1}{d} \sum_{i=1}^{d} W_1^{(s,T)}[r,i] \right)^2. \tag{4}$$

where $W_1^{(s,T)}[r,i]$ denotes the $i$-th element of the $r$-th row of the matrix $W_1^{(s,T)}$.

We first rank experts by the change of router's $l_2$ norm $\Lambda_s^{(T)}$, where those with smaller $\Lambda_s^{(T)}$ are placed higher, which later correspond to higher precision. This ordering is theoretically justified in Section 4, and the intuition is that the model performance is more sensitive to the quantization of those experts $s$ with smaller $\Lambda_s^{(T)}$.

Then, to assign the experts in higher precision that inject high quantization noise to the model, we adjust the ordering by promoting experts with larger maximum intra-neuron variance to higher ranks. Specifically, if a lower-ranked expert $s$ has its $\text{MaxVar}_s^{(T)}$ at least $\zeta$ ($\zeta > 1$)[2] times greater than $\text{MaxVar}_{s'}^{(T)}$ of an expert $s'$, where $s'$ ranks higher than $s$ in the ordering by router norm change, we move $s$ to be above $s'$ in the adjusted ordering. This process is repeated until no further changes are needed. The intuition is that larger intra-neuron variance arises either from wider weight ranges or from more skewed weight concentrations, both of which induce higher quantization noise compared to experts with smaller intra-neuron variance under the same bit-width assignment.

**Special Case**: For pre-trained models where the initial router vectors $w_s^{(0)}$ are unavailable, we use the router's $l_2$ norm itself as a surrogate for $\Lambda_s^{(T)}$, since initial weights are typically small-variance random initializations. $\text{MaxVar}_s^{(T)}$ is computed directly from the pre-trained model as well.

**STEP 2: Bit Assignment**. Based on the obtained ordering, we can assign two-level or three-level quantization as follows.

---

[2] We use $\zeta = 3$ in experiments, since the variance of any bounded distribution is at most three times that of a uniform distribution with the same range. This adjustment affects only 4–5% of experts in experiments.

**Two-level assignment.** Given $b_h > b_l$ and target average bit-width $b_{\text{avg}}$, we quantize the top $\kappa = \frac{b_{\text{avg}} - b_l}{b_h - b_l}$ fraction of experts to $b_h$ and the rest to $b_l$.

**Three-level assignment.** With bit-widths $b_h > b_m > b_l$ and target $b_{\text{avg}}$, we assign higher-ranked experts to $b_h$, mid-ranked experts to $b_m$, and lower-ranked experts to $b_l$. In general, multiple assignment strategies are possible according to the ranking order while still achieving the same $b_{\text{avg}}$. We select the best strategy based on the intuition to balance between maximizing the number of experts assigned to the highest precision and minimizing those assigned to the lowest precision, depending on the value of $b_{\text{avg}}$ relative to the three levels.

Specifically, when $b_{\text{avg}}$ is in $(b_h - (b_h - b_l)/3, b_h)$, i.e., close to $b_h$, we maximize the number of experts assigned to $b_h$. When $b_{\text{avg}}$ is in $[b_h - 2(b_h - b_l)/3, b_h - (b_h - b_l)/3]$, we again maximize the number of experts in $b_h$, but subject to the constraint that the number in $b_l$ does not exceed those in $b_m$. When $b_{\text{avg}}$ is in $(b_l, b_h - 2(b_h - b_l)/3)$, we minimize the number of experts in $b_l$.

# 4 GENERALIZATION GUARANTEES FOR THE ROUTER-NORM-BASED EXPERT-WISE MIXED-PRECISION QUANTIZATION OF MoE

## 4.1 SUMMARY OF THEORETICAL INSIGHTS

Before formally presenting our theoretical setup and results, we first summarize the key theoretical insights. We consider a setting where an MoE model is fine-tuned for a binary classification problem. Each input sequence contains a single task-relevant token that determines the label, while the remaining tokens are task-irrelevant. For each class, there are two distinct task-relevant tokens: one is more prevalent, appearing in a $(1 - \alpha)$ fraction of the data, while the other appears in an $\alpha$ fraction ($\alpha < \frac{1}{4}$). Although based on the simplified setup, our theoretical insights are validated empirically on practical MoE models in different language tasks. Our major theoretical takeaways include:

**1. Experts specialized in learning less-prevalent tokens undergo smaller changes in their router's $l_2$ norm than experts that learn more-prevalent tokens.** We show that different experts specialize in different task-relevant tokens. The routers associated with experts that exclusively learn the less-prevalent token exhibit a smaller $l_2$ norm change after fine-tuning, compared to routers for experts that learn more-prevalent tokens. This observation suggests that the router norm change can serve as a useful indicator for distinguishing between these two types of experts.

**2. Experts that learn less-prevalent tokens produce weaker activations, and the model's generalization performance is more sensitive to the quantization of these experts.** We prove that experts are primarily activated by the task-relevant tokens they learn. Experts that learn less-prevalent tokens generate significantly weaker activations than experts that learn more-prevalent tokens. Therefore, the model performance is more sensitive to the quantization of the former ones.

**3. Quantizing experts with smaller router $\ell_2$ norm changes to higher precision, while rest of the experts in lower precision achieves the same generalization as full-precision quantization.** Because the model's generalization is more sensitive to the quantization of the experts learning less-prevalent tokens, and as they can be identified via router's $\ell_2$ norm change, quantizing them to $b_h$ allows safe reduction of other experts' precision by $\log_2 \left( \frac{1-\alpha}{\alpha} \right)$ bits without hurting generalization.

## 4.2 DATA MODEL AND ASSUMPTIONS

**The MoE model and binary classification task.** We consider a neural network that contains a single MoE block, fine-tuned on a binary supervised classification task, where each input sequence $x$ is labeled with $y \in \{+1, -1\}$. The MoE block generates one-dimensional output tokens, i.e., $d' = 1$ for $x_{out}^{(j)}$ in (1), and the model output is computed by aggregating all the output tokens, i.e., for an input sequence $x$, the model's output is

$$f(x) := \sum_{j \in [n]} x_{out}^{(j)} = \sum_{j \in [n]} \sum_{s \in [k]} f_s(x^{(j)}) \tag{5}$$

Let $f^{(T)}(\cdot)$ denote the model after $T$ steps of finetuning. $x$ is correctly classified if $yf^{(T)}(x) > 0$. For each expert $s \in [k]$, the second layer weights are fixed[3] during training and defined as $W_2^{(s)} := a^{(s)} \cdot 1^{1 \times m}$, where $a^{(s)} \in \{+1, -1\}$. We refer to each expert as positively connected to the final output if $a^{(s)} = 1$, and negatively connected if $a^{(s)} = -1$. Let $S_+, S_- \subset [k]$ denote the set of positively and negatively connected experts, respectively. The activation function $\sigma(\cdot)$ is rectified linear unit (ReLU). The routing mechanism follows expert-choice routing, where each expert selects $l$ tokens, satisfying $l \leq L$ for some constant $L$.

Although our theoretical analysis is based on a two-layer MoE model, it already captures the key components, including routers, experts, and nonlinear activation, and the learning problem is already highly non-convex. In fact, the two-layer network model is SOTA for theoretical analysis of training dynamics and generalization in MoEs (Chen et al., 2022b; Chowdhury et al., 2023), and in general deep neural networks (Li et al., 2023; Zhang et al., 2023; Allen-Zhu & Li, 2023; Bu et al., 2024).

**Two-precision-level quantization**. To simplify the theoretical analysis, we consider two precision levels and only the first layer weights of each expert $s$, i.e., $W_1^{(s,T)}$, are quantized. The top $\kappa$-fraction of experts in $S_+$ and the top $\kappa$-fraction in $S_-$, each with the smallest values of $\Lambda_s^{(T)}$, are quantized to the higher bit-width $b_h$, while the remaining experts in both sets are quantized to the lower bit-width $b_l$. The quantization is applied in a column-wise fashion: for each expert $s$ and its corresponding bit-width, the bin size $\Delta$ is computed independently for each column of $W_1^{(s,T)}$. Without loss of generality, we assume the zero-point $z = 0$.

**The data model**. Let $\mathcal{P} \subset \mathbb{R}^d$ denote a set of orthonormal vectors with $|\mathcal{P}| \leq d = \Omega(L^8)$. Two vectors $o_1$ and $o_2$ in $\mathcal{P}$, and their negatives $-o_1$ and $-o_2$ are called *task relevant*, denoted by set $\mathcal{P}_r = \{\pm o_1, \pm o_2\}$, while all vectors in $\mathcal{P} \backslash \{o_1, o_2\}$ are *task-irrelevant*.

Each sequence and label pair $(x, y)$ follows a distribution $\mathcal{D}$, where $x$ contains exactly one token from $\mathcal{P}_r$, which determines $y$: sequences containing $\pm o_1$ are labeled as class 1 (i.e., $y = +1$), and those containing $\pm o_2$ are labeled as class 2 (i.e., $y = -1$). The remaining tokens in $x$ are drawn independently from the task-irrelevant set $\mathcal{P} \backslash \{o_1, o_2\}$, each with probability $O(1/d)$. With probability $\alpha$, where the constant $\alpha$ is in $(0, 1/4)$, a sequence contains the **less prevalent** task-relevant tokens $o_1$ or $o_2$, and with probability $1 - \alpha$, it contains the **more prevalent** tokens $-o_1$ or $-o_2$.

Our data model is similar to Bu et al. (2024) except that our task-irrelevant vectors are drawn from an orthonormal set instead of a Gaussian distribution. The assumption of orthonormal task-irrelevant vectors have been widely deployed in theoretical analysis (Brutzkus & Globerson, 2021; Shi et al., 2022; Allen-Zhu & Li, 2022; Chen et al., 2022b; Zhang et al., 2023; Li et al., 2023).

We next introduce some useful notations in presenting our theoretical results. After $t$ training iterations, we define the activation of expert $s$ in response to a task-relevant token as follows:

**Definition 4.1.** *For expert $s \in [k]$, the activation of expert $s$ by a task-relevant vector is defined as*

$$\sigma_v^{(s,t)} := \vec{1}^\top \sigma(W_1^{(s,t)\top} v), \quad v \in \mathcal{P}_r, \tag{6}$$

*where $W_1^{(s,t)}$ is the first-layer weights of expert $s$ at iteration $t$, and $\vec{1}$ is an all-ones vector in $\mathbb{R}^m$.*

Intuitively, a lower activation of an expert for a task-relevant vector leads to a smaller gap between the output of this expert and the output of another expert not selecting the token, leading to weaker predictions against quantization noise.

The same as Chowdhury et al. (2024), we define an expert's *proficiency measure* to quantify the router's ability to select task-relevant tokens from a sequence. Specifically,

**Definition 4.2.** *The proficiency of expert $s$ after $t$ training iterations in selecting a task-relevant vector $v$ is measured by the probability that it assigns a gating value of at least $1/l$ to token $v$, i.e.,*

$$p_v^{(s,t)} := \mathbb{P}[G_j^{(s,t)} \geq 1/l \mid x^{(j)} = v \text{ for some } j \in [n]], \quad v \in \mathcal{P}_r \tag{7}$$

**Alignment of the pretrained model**. In a pretrained model $(t = 0)$, we say the router for expert $s$ is *aligned* to task-relevant vector $v$ ($v \in \mathcal{P}_r$) if $p_v^{(s,0)} = \Omega(1)$, i.e., it selects $v$ with a nontrivial gating

---

[3]Fixing the output layer for analytical convenience is standard in the literature and has been adopted in prior works (Li & Liang, 2018; Brutzkus et al., 2018; Arora et al., 2019; Zhang et al., 2023; Chowdhury et al., 2023)

value for a constant fraction of samples containing $v$. We assume that in the pretrained model, routers of the experts in $S_+$ are aligned to either $o_1$ or $-o_1$, and routers of the experts in $S_-$ are aligned to either $o_2$ or $-o_2$. This assumption reflect the common intuition that a pretrained MoE model learns to specialize experts for different subtasks or feature types,

Let $S_v$ ($v \in \mathcal{P}_r$) denote the set of experts whose routers are aligned with $v$ in the pretrained model. Let $\gamma = \max(\frac{|S_{o_1}|}{|S_+|}, \frac{|S_{o_2}|}{|S_-|})$ denote the maximum fraction of routers that are aligned to less-prevalent task-relevant vectors in $S_+$ and $S_-$.

## 4.3 MAIN THEORETICAL RESULTS

**Lemma 4.3.** *Suppose the pretrained model is fine-tuned for $T = \Theta(l^2\sqrt{\log l}/\alpha)$ iterations, the returned $f^{(T)}$ has the following properties,*

*(i) the routers' alignment to task-relevant vectors are enhanced during training, specifically,*

$$p_v^{(s,T)} = 1, \ p_{-v}^{(s,T)} = 0, \quad \forall s \in S_v, \forall v \in \mathcal{P}_r = \{\pm o_1, \pm o_2\} \tag{8}$$

*(ii) the $l_2$ norm change of the routers aligned with the more prevalent $-o_1$ and $-o_2$ are higher than those aligned with the less prevalent $o_1$ and $o_2$, specifically,*

$$\Lambda_{s'}^{(T)} > \Lambda_s^{(T)}, \quad \forall s \in S_{o_i}, \forall s' \in S_{-o_i}, i = 1, 2 \tag{9}$$

*and (iii) the expert activation by $-o_1$ and $-o_2$ are higher than that by $o_1$ and $o_2$, specifically,*

$$\sigma_{o_i}^{(s,T)} = \Omega(ml\sqrt{\log l}), \quad \sigma_{-o_i}^{(s',T)} = \Omega\left(\frac{(1-\alpha)}{\alpha}ml\sqrt{\log l}\right), \quad \frac{\sigma_{-o_i}^{(s',T)}}{\sigma_{o_i}^{(s,T)}} \geq \frac{1-2\alpha}{2\alpha} \tag{10}$$

Lemma 4.3 summarizes key properties of the fine-tuned model that can be leveraged for expert-wise mixed-precision implementation. First, (8) shows that if the router of an expert $s$ is aligned with a task-relevant vector $v$ in the pretrained model, that is, $p_v^{(s,0)} = \Omega(1)$, then after fine-tuning, the alignment becomes stronger: $p_v^{(s,T)} = 1$. Moreover, the expert $s$ suppresses $-v$ by assigning zero or negligible gating value to the negative token $-v$, i.e., $p_{-v}^{(s,T)} = 0$. This implies that each expert becomes specialized in a single task-relevant vector after fine-tuning.

Second, (9) shows that the change in the router's $l_2$-norm is larger for experts aligned with more prevalent features, and smaller for those aligned with less prevalent ones. This property allows us to distinguish two types of experts based on their router norm changes. Third, (10) demonstrates that experts aligned with less prevalent vectors produce weaker activations than those aligned with more prevalent ones, resulting from the less frequent occurrence of these tokens in the data. The ratio of their activations is at least $(1 - 2\alpha)/(2\alpha)$. Thus, the model's generalization is expected to be more sensitive to the quantization of the experts aligning with less prevalent vector, and hence these experts need higher precision. Lemma 4.3 is verified by synthetic data in Figs. 4 and 5 in Appendix A.

We next formally establish Theorem 4.4 that characterizes the generalization guarantee of the quantized model $f_Q^{(T)}$ after applying the two-level mixed-precision quantization method in Section 3.3 to the fine-tuned model $f^{(T)}$. Let $\text{Var}_r^{(s,T)}$ denote the variance of the $r$-th column of $W_1^{(s,T)}$.

**Theorem 4.4.** *Suppose the number of fine-tuning iterations satisfies $T = \Theta(l^2\sqrt{\log l}/\alpha)$, and $\max_{r \in [m]} \text{Var}_r^{(s,T)} = \Theta(1)$ for every expert $s$. If $\kappa \geq \gamma$, and the two quantization levels satisfy*

$$b_h \geq \log_2(1 + \Omega(d\sqrt{\log(kmd^2)/l^2\log l})) \tag{11}$$

*and*

$$b_l \geq \log_2(1 + \frac{\alpha}{1-\alpha}\Omega(d\sqrt{\log(kmd^2)/l^2\log l})), \tag{12}$$

*then with high probability the quantized model has guaranteed generalization, i.e.,*

$$\mathbb{P}[\forall(x,y) \sim \mathcal{D} : yf_Q^{(T)}(x) > 0] = 1. \tag{13}$$

*Remark* 4.5. Theorem 4.4 states that if the maximum intra-neuron variance of all the experts are close to each other, i.e., $\forall s \in [k], \mathrm{MaxVar}_s^{(T)} = \Theta(1)$, sorting the experts in ascending order of their router norm change $\Lambda_s^{(T)}$, and quantizing the top $\kappa$-fraction ($\kappa \geq \gamma$) of experts in this sorted list to $b_h$ bits and the remaining experts to $b_l$ bits, where $b_h$ and $b_l$ satisfy conditions (11) and (12), allows the quantized model to preserve the generalization of the full-precision model. Each low-precision expert can use $\log_2 \left( \frac{1-\alpha}{\alpha} \right)$ fewer bits than its high-precision counterpart. This is verified by synthetic data in Fig. 6 in Appendix A. Note that all the experts aligned with less prevalent vectors are among the top $\kappa$-fraction (by Lemma 4.3 (ii)) and exhibit smaller activation values (by Lemma 4.3 (iii)). Therefore, they require higher precision. In contrast, the experts quantized to lower precision are those aligned with more prevalent vectors and have larger activations, and hence can be quantized aggressively.

## 5 EXPERIMENTAL RESULTS

### 5.1 EXPERIMENTS ON FINETUNED SWITCH-TRANSFORMER MODEL

Here, we present quantization results on Switch Transformer (Fedus et al., 2022) finetuned on CNN/Daily Mail (CNNDM) text summarization task (See et al., 2017). All non-MoE weights are quantized to 8 bits. We apply HQQ (Badri & Shaji, 2023) for quantizing the model[4]. See section B in appendix for more implementation details.

**Two-level expert-wise mixed-precision**: As shown in Figure 2, uniform 3-bit expert quantization nearly preserves generalization, while uniform 2-bit severely degrades the generalization. We therefore use mixed-precision with two bit levels: 3 and 2.

**Our method outperforms existing expert-wise mixed-precision methods**: We benchmark against two prior expert-wise mixed-precision strategies: (i) activation frequency (average tokens routed per expert) and (ii) activation weights (average gating weights on a calibration set) (Li et al., 2024; Huang et al., 2025a), where higher-frequency/weight experts are assigned higher precision. In contrast, our router-norm-change ordering itself preserves generalization down to 2.5 average bits/expert, outperforming both baselines. Furthermore, additional reordering by $\mathrm{MaxVar}$ affects only 3.7% of experts, extending preservation to 2.125 bits.

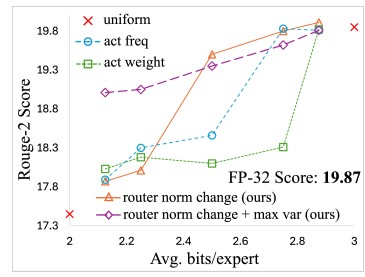

Figure 2: Expert-wise mixed-precision of Switch Transformer on CNNDM. Bit choices: $2, 3$

### 5.2 EXPERIMENTS ON PRETRAINED MIXTRAL MODELS

We quantize the pretrained Mixtral 8x7B (46.7B parameters) and Mixtral 8x22B (140.6B parameters) models (Jiang et al., 2024) using GPTQ (Frantar et al., 2023) to evaluate on eight zero-shot benchmark LLM tasks. All non-MoE parameters are assigned to 3 bits. See section B in the appendix for details.

**Baselines**. We compare against the state-of-the-art expert-wise method, *Pre-loading Mixed-precision Quantization* (PMQ) (Huang et al., 2025a), which assigns bit-widths by minimizing Frobenius-norm output errors (per expert and bit level) weighted by activation frequency and gating scores. As PMQ outperforms the activation frequency and activation weights based methods, the comparison with these method are in Appendix (see Figure 7, 8 in Appendix). We also evaluate non-expert-wise approaches, including layer-wise (Hessian (Dong et al., 2020), BSP (Li et al., 2024)) and group-wise (Slim-LLM (Huang et al., 2025b)) methods. Our method has advantages as follows.

**Three-level expert-wise mixed precision**. We consider three-level bit-assignment of (1,2,3) bits. As shown in Table 1, the uniform 3-bit expert quantization almost maintains generalization, but uniform 2-bit quantization significantly degrades performance.

**High accuracy with robust scaling**. Our method surpasses PMQ above 2.0 average bits in terms of accuracy for Mixtral 8x7B[5]. Extending to Mixtral-8x22B (140.6B parameters), our method

---

[4]We use eight V100 GPUs for fine-tuning and one NVIDIA A5000 GPU (48GB) for quantized inference.

[5]Compressing below 2.0 bits/expert is too aggressive, since the compressed model size ($\leq$13.1 GB) falls below the equivalent dense model (13.6 GB) that is trained with far more data and has better generalization.

again outperforms PMQ, demonstrating robustness to model scale. It also outperforms non-expert-wise methods (e.g., Hessian (layer-wise) (Dong et al., 2020), BSP (layer-wise) (Li et al., 2024), and Slim-LLM (group-wise) (Huang et al., 2025b)) by large margins.

**Low inference cost**. Figure 3 shows inference time on Wikitext2 (Merity et al., 2016). For the same average bits/expert, our method is faster than PMQ because PMQ assigns higher precision to frequently activated experts, whereas our method allocates higher precision to less frequent experts, reducing computation.

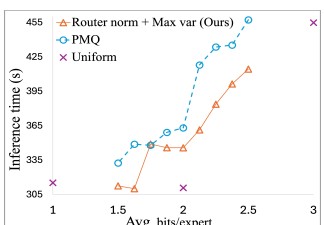

**Negligible assignment overhead**. Unlike PMQ, which requires evaluating all experts across bit levels on a calibration set (e.g., 110 GB GPU memory and 2227s for Mixtral-8x7B; 350 GB and 6000s for Mixtral-8x22B), our method only sorts experts by router norm with minor reordering. This requires no GPU and negligible computation, enabling scalable compression of large MoE models.

Figure 3: Inference time of different expert-wise methods.

Table 1: Task-wise accuracy (%) of different methods on the 8 benchmark LLM tasks.

| Model | Method | Avg. bits/exp. | Memory (GB) | PIQA | ARC-e | ARC-c | BoolQ | HellaS. | Wino. | MathQA | MMLU | Avg. |
|---|---|---|---|---|---|---|---|---|---|---|---|---|
| Mixtral 8x7B | Full-precision | 16 (FP) | 96.8 | 83.68 | 83.50 | 59.64 | 85.05 | 83.99 | 76.4 | 41.61 | 67.85 | 72.72 |
| | Uniform | 3 | 19.3 | 82.32 | 80.05 | 57.42 | 86.09 | 81.51 | 75.14 | 39.43 | 64.84 | 70.85 |
| | | 2 | 13.1 | 76.44 | 67.68 | 43.60 | 72.51 | 72.93 | 65.27 | 28.58 | 42.79 | 58.73 |
| | Router norm + Max var (**Ours**) | 2.75 | 17.7 | 81.83 | **80.47** | 56.31 | **85.57** | 81.05 | 74.98 | 38.29 | 61.55 | **70.01** |
| | | 2.625 | 16.9 | 81.45 | **78.62** | 54.86 | 85.60 | 80.57 | **74.66** | 36.75 | **59.78** | **69.04** |
| | | 2.5 | 16.1 | **80.79** | **78.41** | 54.44 | **85.14** | 79.36 | 74.35 | 36.28 | 58.23 | **68.38** |
| | | 2.375 | 15.3 | 80.20 | 75.38 | 51.19 | 84.92 | 78.69 | 73.80 | 33.47 | **56.07** | **66.72** |
| | | 2.25 | 14.5 | **80.41** | 72.90 | 50.09 | 84.04 | 77.46 | 73.95 | 31.96 | 55.53 | 65.79 |
| | | 2.125 | 13.8 | 78.94 | 74.45 | 51.02 | 80.12 | 76.56 | 70.17 | 31.29 | **51.56** | 64.26 |
| | | 2.0 | 13.1 | **77.26** | 71.17 | 46.84 | 80.61 | 74.17 | 69.93 | 30.18 | 50.34 | 62.56 |
| | | 1.75 | 11.7 | 75.03 | **69.53** | 42.92 | 73.64 | 70.03 | 68.35 | 27.04 | 44.82 | 58.95 |
| | PMQ | 2.75 | 17.7 | **82.05** | 78.87 | **56.48** | 84.80 | **81.15** | 75.30 | 38.39 | **61.79** | 69.85 |
| | | 2.625 | 16.9 | **81.56** | 78.41 | 52.99 | 83.67 | 80.04 | 74.66 | **38.26** | 59.76 | 68.67 |
| | | 2.5 | 16.1 | 80.63 | 76.94 | 53.33 | 83.15 | **80.02** | **74.98** | **37.15** | 54.05 | 67.53 |
| | | 2.375 | 15.3 | **80.47** | 73.32 | 50.00 | 81.93 | 78.54 | **74.66** | **35.18** | 53.34 | 65.93 |
| | | 2.25 | 14.5 | 80.14 | 72.14 | 49.32 | 83.15 | **77.62** | **74.19** | **33.70** | 51.00 | 65.16 |
| | | 2.125 | 13.8 | **79.00** | **75.51** | 49.91 | 72.26 | **76.76** | **72.30** | **34.07** | 50.53 | 63.79 |
| | | 2.0 | 13.1 | 76.93 | **71.93** | 46.59 | 78.65 | **74.88** | **73.24** | **31.83** | 48.60 | **62.83** |
| | | 1.75 | 11.7 | **76.66** | 69.19 | **44.28** | **79.63** | 70.85 | **71.19** | **29.88** | 42.52 | **60.53** |
| | Hessian | 2.5 | 17.0 | 80.21 | 76.38 | 51.20 | 81.11 | 78.05 | 72.97 | 35.27 | 56.21 | 67.18 |
| | | 2.25 | 15.3 | 79.21 | 72.41 | 46.70 | 79.15 | 76.38 | 71.25 | 31.97 | 50.60 | 63.47 |
| | | 2.0 | 13.6 | 75.32 | 67.26 | 45.01 | 70.29 | 71.90 | 69.11 | 31.07 | 40.85 | 58.85 |
| | BSP | 2.5 | 17.0 | 68.23 | 54.97 | 28.38 | 68.16 | 55.61 | 62.19 | 24.07 | 27.74 | 49.07 |
| | Slim-LLM | 2.0 | 13.6 | 61.70 | 49.07 | 28.24 | 66.18 | 44.10 | 57.54 | 23.62 | 25.43 | 44.49 |
| Mixtral 8x22B | Full-precision | 16 (FP) | 281.2 | 85.12 | 84.01 | 60.12 | 86.23 | 84.50 | 77.40 | 42.10 | 68.20 | 76.31 |
| | Uniform | 3 | 57.5 | 81.45 | 76.68 | 53.07 | 78.53 | 74.23 | 68.19 | 36.21 | 55.46 | 65.48 |
| | | 2 | 38.6 | 55.98 | 31.31 | 22.78 | 57.92 | 29.23 | 50.12 | 21.64 | 23.24 | 36.53 |
| | Router norm +Max var (**Ours**) | 2.5 | 46.7 | **80.14** | **71.25** | **47.27** | 73.49 | 65.11 | 64.40 | **30.62** | 48.16 | 60.10 |
| | | 2.25 | 43.0 | **79.27** | **69.61** | **46.25** | **72.23** | 64.75 | **65.43** | **29.45** | **46.33** | **59.17** |
| | | 2.0 | 38.6 | **78.56** | 64.18 | **41.30** | 68.26 | **61.91** | 61.25 | 27.14 | 40.09 | 55.34 |
| | | 1.75 | 35.2 | **75.14** | **63.22** | **37.12** | 65.96 | **52.95** | 59.59 | 25.19 | 32.36 | **51.44** |
| | PMQ | 2.5 | 46.7 | 79.49 | 70.62 | 46.84 | **74.28** | **68.64** | **65.35** | 29.88 | **50.40** | **60.69** |
| | | 2.25 | 43.0 | 78.78 | 69.15 | 42.41 | 55.14 | 62.29 | 61.96 | 28.91 | 44.16 | 55.35 |
| | | 2.0 | 38.6 | 76.55 | **66.16** | 39.85 | **69.48** | 60.11 | 59.83 | 27.00 | 39.43 | 54.80 |
| | | 1.75 | 35.2 | 72.20 | 56.90 | 32.59 | 59.33 | 49.43 | 57.38 | 24.82 | 30.30 | 47.87 |

## 5.3 ABLATION STUDY

We determine the importance of the two stages of the experts' ranking: the *router norm* based ranking and the *maximum intra-neuron variance* (i.e., $\mathrm{MaxVar}$) based reordering by providing an ablation study among (i) only $\mathrm{MaxVar}$ based ranking, (ii) only *router norm* based ranking, and (ii) *router norm* based ranking + $\mathrm{MaxVar}$ based reordering described in section 3.3. We conduct the study on Mixtral 8x7B for the eight zero-shot benchmark LLM tasks. We report the average accuracy across the tasks for both the two-level assignment (bit choices: 2, 3), and three-level assignment (bit choices: 1, 2, 3) in Table 2. As we can see, for the two-level assignment, only router norm based ranking performs better than only $\mathrm{MaxVar}$ based ranking for most of the average bit points. However, for the

three-level assignment, there is an abrupt drop in performance in the only router-norm-based ranking as some of the unusually large MaxVar experts are placed in 1 bit (see our MaxVar visualization of Mixtral 8x7B in Appendix E), which injects an unbearable amount of quantization noise into the model. Reordering these experts (only 11 out of 256 for $\zeta = 3$) to higher rank completely removes this issue and significantly outperforms the only MaxVar based method and other competitive baselines provided in Table 1 of the paper. We provide an empirical justification for our selection of $\zeta$ in Appendix C.

Table 2: Average accuracy (%) of different expert ranking methods

| Method | Two-level assignment (bit choices: 2, 3) | | | | | | Three-level assignment (bit choices: 1, 2, 3) | | | | |
|---|---|---|---|---|---|---|---|---|---|---|---|
| | Avg. bits/expert | | | | | | Avg. bits/expert | | | | |
| | 2.75 | 2.625 | 2.5 | 2.375 | 2.25 | 2.125 | 2.75 | 2.5 | 2.25 | 2.0 | 1.75 |
| MaxVar | 69.51 | **68.51** | 66.01 | 64.24 | 63.88 | 60.65 | 69.37 | 67.90 | 63.97 | 60.44 | 58.11 |
| Router norm | **69.92** | 68.35 | 67.01 | 64.97 | 63.84 | **61.54** | 54.23 | 49.78 | 48.12 | 44.96 | 42.92 |
| Router norm + MaxVar (Our method) | 69.50 | 68.40 | **67.17** | **65.31** | **64.26** | 61.43 | **70.01** | **68.38** | **65.79** | **62.56** | **58.95** |

## 5.4 JUSTIFICATION FOR USING FINAL ROUTER NORM AS A SURROGATE FOR CHANGE IN NORM OF PRETRAINED MoE MODELS

As stated in section 3.3, for the experiments on zero-shot evaluation of pre-trained models, we propose to use the final router norm ($w_s^{(T)}$) to approximate the change in the router's norm ($\Lambda_s^{(T)}$), when the randomly initialized model is not publicly available for computing the initial router norm ($w_s^{(0)}$). The rationale behind the approximation comes from the fact that the initial routers are generally initialized randomly with small variance (e.g., parameters of DeepSeekMoE are initialized randomly with variance 0.000036 (Dai et al., 2024)), which leads to a very small difference between the two methods. We provide a theoretical justification of our claim in Appendix G. Here, we provide an empirical justification for the claim by reinitializing the routers of the pre-trained switch transformer randomly from $\mathcal{N}(0, \sigma^2)$ with $\sigma = 0.0005$. We finetune the re-initialized model on the CNN/Daily Mail dataset and compare the rank correlation between the two expert ranking methods measured via Spearman's $\rho$ and Kendall's $\tau$ ($\rho, \tau \approx 1$ implies high correlation). The results are provided in Table 3. The high rank correlation implies that the rank orders using both methods are very similar.

Table 3: Correlation between final router norm and change in norm across different layers

| | Enc-1 | Enc-3 | Enc-5 | Enc-7 | Enc-9 | Enc-11 | Dec-1 | Dec-3 | Dec-5 | Dec-7 | Dec-9 | Dec-11 |
|---|---|---|---|---|---|---|---|---|---|---|---|---|
| Spearman's $\rho$ | 0.9997 | 0.9994 | 0.9995 | 0.9992 | 0.9989 | 0.9990 | 0.9990 | 0.9997 | 0.9995 | 0.9997 | 0.9998 | 0.9999 |
| Kendall's $\tau$ | 0.9950 | 0.9900 | 0.9920 | 0.9871 | 0.9851 | 0.9861 | 0.9871 | 0.9960 | 0.9920 | 0.9950 | 0.9960 | 0.9980 |

We provide the quantization results for both methods in Table 4. As expected, due to the high correlation of the rank order between the final norm and the change in norm based method, the scores of both methods are very similar.

Table 4: Quantization results for final router norm and change in router norm based method

| Initial router | Original pretrained | Random router | | | | | | |
|---|---|---|---|---|---|---|---|---|
| Method | Full-precision | Full-precision | Change in norm | | | Final norm | | |
| Avg. bits/expert | 32 (FP) | 32 (FP) | 2.75 | 2.5 | 2.25 | 2.75 | 2.5 | 2.25 |
| Rouge-2 score | 19.87 | 19.46 | 18.79 | 18.60 | 18.37 | 18.81 | 18.59 | 18.38 |

## 6 CONCLUSION

This paper proposes an expert-wise mixed-precision quantization method for MoE models that allocates higher bit-widths to experts with smaller router norms and lower bit-widths otherwise. It can use pretrained router norms as an alterative to avoid costly fine-tuning while maintaining accuracy. The approach is theoretically supported and empirically effective on large MoE models. It reduces memory and inference costs, promoting energy efficiency and a smaller carbon footprint. Future work will combine the method to layer-wise and block-wise quantization.

## ACKNOWLEDGMENTS

This work was supported in part by the 2024 IBM PhD fellowship, in part by the National Science Foundation (NSF) under Grant 2430223, in part by Army Research Office (ARO) under Grant W911NF-25-1-0020, and in part by the RPI-IBM Future of Computing Research Collaboration (http://airc.rpi.edu), part of the IBM AI Horizons Network (http://ibm.biz/AIHorizons).

## REPRODUCIBILITY STATEMENT

We provide the complete setup for our theoretical analysis in section 4.2. We include additional details related to our analysis in Appendix H. We provide the proof of Lemma 4.3 in Appendix I, and the proof of Theorem 4.4 in Appendix K. We provide the details of our experimental setup, including model architecture, parameter size, values of the hyperparameters related to the implemented quantization methods, and the evaluation datasets in Appendix B. We include the code of our experiments for reproducibility.

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

# A   VERIFICATION OF THEORETICAL RESULTS ON SYNTHETIC DATA

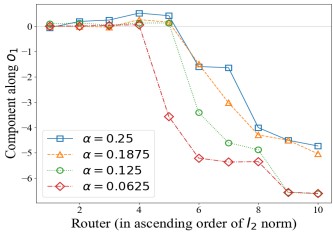

Figure 4: Router vector projection along the class-1 task-relevant feature $o_1$.

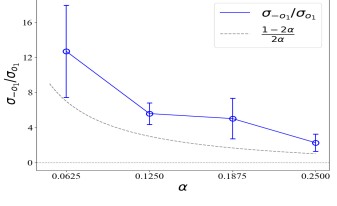

Figure 5: Ratio of activations of experts learning more and less prevalent tokens, respectively (class-1)

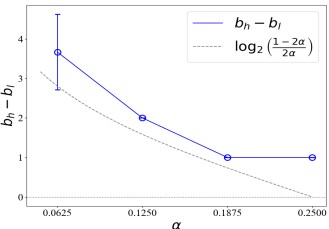

Figure 6: Change of bit difference between higher bit experts and lower bit experts with $\alpha$

We validate our theoretical claims using synthetic data generated as described in Section 4.2. Tokens are drawn from an orthonormal matrix obtained via QR decomposition of a $d \times d$ Gaussian matrix with $d = 200$. We set $k = 20$, $m = 800$, $n = 100$, and $l = 5$. Model weights are initialized from a zero-mean Gaussian distribution with variance $0.0001$ and trained with a learning rate of $0.2$.

Figure 4 shows the projection of the router vectors onto the direction of $O_1$, a class-1 task-relevant vector, for the experts in $S_+$. The experts are sorted in ascending order based on the change in router norm. Routers exhibiting larger norm changes tend to have a significant component along the more dominant $-O_1$ direction, consistent with Lemma 4.3(ii). Figure 5 presents the minimum ratio of activation of $-O_1$-aligned experts by $-o_1$ to that of $O_1$-aligned experts by $o_1$, minimized over all such expert pairs. This ratio is compared against the theoretical lower bound of $(1 - 2\alpha)/2\alpha$, as established in Lemma 4.3(iii).

We quantize the weights as described in Section 4.2, using equation (2). Experts with large components along the $-o_1$ and $-o_2$ directions are quantized to a lower bit-width $b_l$, unless a high maximum row variance is observed, in which case they are quantized to a higher bit-width $b_h$. The value of $b_h$ is determined empirically as the minimum bit-width required to achieve zero test error when all experts are uniformly quantized to $b_h$. We then choose $b_l$ as the minimum bit-width that still maintains zero test error in the mixed-precision setting. Figure 6 shows that the gap $b_h - b_l$ increases as $\alpha$ decreases, aligning with the theoretical bound $\log_2((1 - 2\alpha)/2\alpha)$ discussed in Remark 4.5.

# B   DETAILS ON THE QUANTIZED MODELS AND EVALUATION TASKS

## B.1   SWITCH TRANSFORMER

We fine-tune a pre-trained Switch Transformer (Fedus et al., 2022), which contains 64 experts per MoE block on CNN/Daily Mail (CNNDM) text summarization task (See et al., 2017). The model follows an encoder-decoder architecture with 12 transformer blocks each in the encoder and decoder; every even-numbered block is an MoE block, resulting in 12 MoE blocks total. The model has about 2 billion parameters, with 90% residing in MoE blocks. All non-MoE weights are quantized to 8 bits. We apply HQQ (Badri & Shaji, 2023) for quantizing the model. Weights in each row of the weight matrices are quantized together.

## B.2   MIXTRAL

We quantize the pretrained Mixtral 8x7B and Mixtral 8x22B models (Jiang et al., 2024), which adopt a decoder-only architecture. The Mixtral 8x7B contains 32 transformer blocks, and the Mixtral 8x22B contains 56 transformer blocks. All blocks are MoE blocks of 8 experts. Mixtral 8x7B contains 46.7B parameters, with 97% residing in the MoE blocks. Mixtral 8x22B contains 140.6B parameters, with 99% residing in the MoE blocks. We quantize the non-MoE parameters to 4 bits and apply GPTQ (Frantar et al., 2023) for model quantization with group size 128, and 1% damping. We use 128 samples of length 2048 from Wikitext2 as the GPTQ calibration data. Model performance is evaluated on eight zero-shot benchmark LLM tasks: PIQA (Bisk et al., 2020), ARC-Challenge and ARC-Easy (Clark et al., 2018), BoolQ (Clark et al., 2019), HellaSwag (Zellers et al., 2019),

WinoGrande (Keisuke et al., 2019), MathQA (Amini et al., 2019), and MMLU (Hendrycks et al., 2021), using the EleutherAI LM Harness (Gu et al., 2024).

## C  JUSTIFICATION FOR THE SELECTION OF $\zeta$

As stated in section 3.3, we select $\zeta = 3$, since the variance of any bounded distribution is at most three times that of a uniform distribution with the same range. Our selection of $\zeta$ alters the initial router norm based order by a very small amount (only 11 out of 256 experts are reordered in Mixtral 8x7B, and only 28 out of 768 experts are reordered in Switch Transformer for our selection of $\zeta$). Indeed, our selection of $\zeta$ only reorders the experts that have unusually large $\mathrm{MaxVar}$ values in an MoE layer (see our $\mathrm{MaxVar}$ visualization of Mixtral 8x7B in Appendix E). We conduct a sweep of $\zeta$ for Mixtral 8x7B on the eight benchmark downstream tasks and report the average accuracy in Table 5. As we can see, the performance picks around $\zeta = 3.0$, which justifies our selection.

Table 5: Avg. accuracy (%) for different values of $\zeta$

| Avg. bits/expert | $\zeta$ | | | | | |
| --- | --- | --- | --- | --- | --- | --- |
| | 1.0 | 2.0 | 2.5 | 3.0 | 4.0 | 5.0 |
| 2.0 | 60.44 | 61.56 | 62.28 | **62.56** | 61.32 | 61.74 |

## D  MORE RESULTS ON MIXTRAL 8X7B

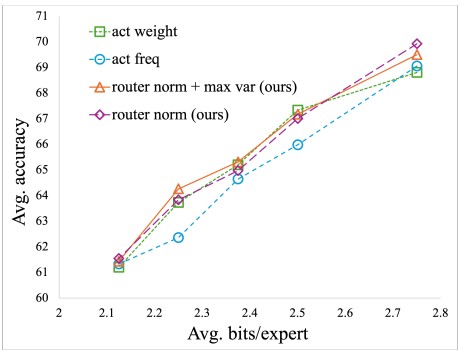

Figure 7: Expert-wise mixed-precision results for Mixtral 8x7B on eight benchmark LLM tasks; expert bit-choices: $2, 3$. Only 4.3% of the experts are reordered to higher ranks in maximum intra-neuron based reordering.

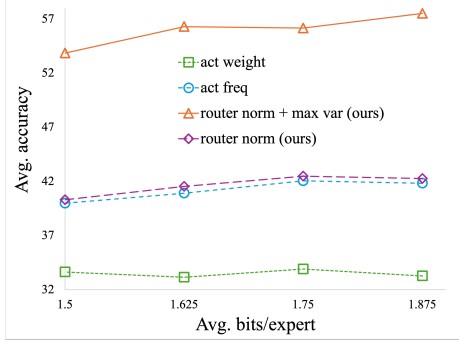

Figure 8: Expert-wise mixed-precision results for Mixtral 8x7B on eight benchmark LLM tasks; expert bit-choices: $1, 2$. Only 4.3% of the experts are reordered to higher ranks in maximum intra-neuron based reordering.

# E    VISUALIZATION OF MaxVar FOR MIXTRAL 8X7B

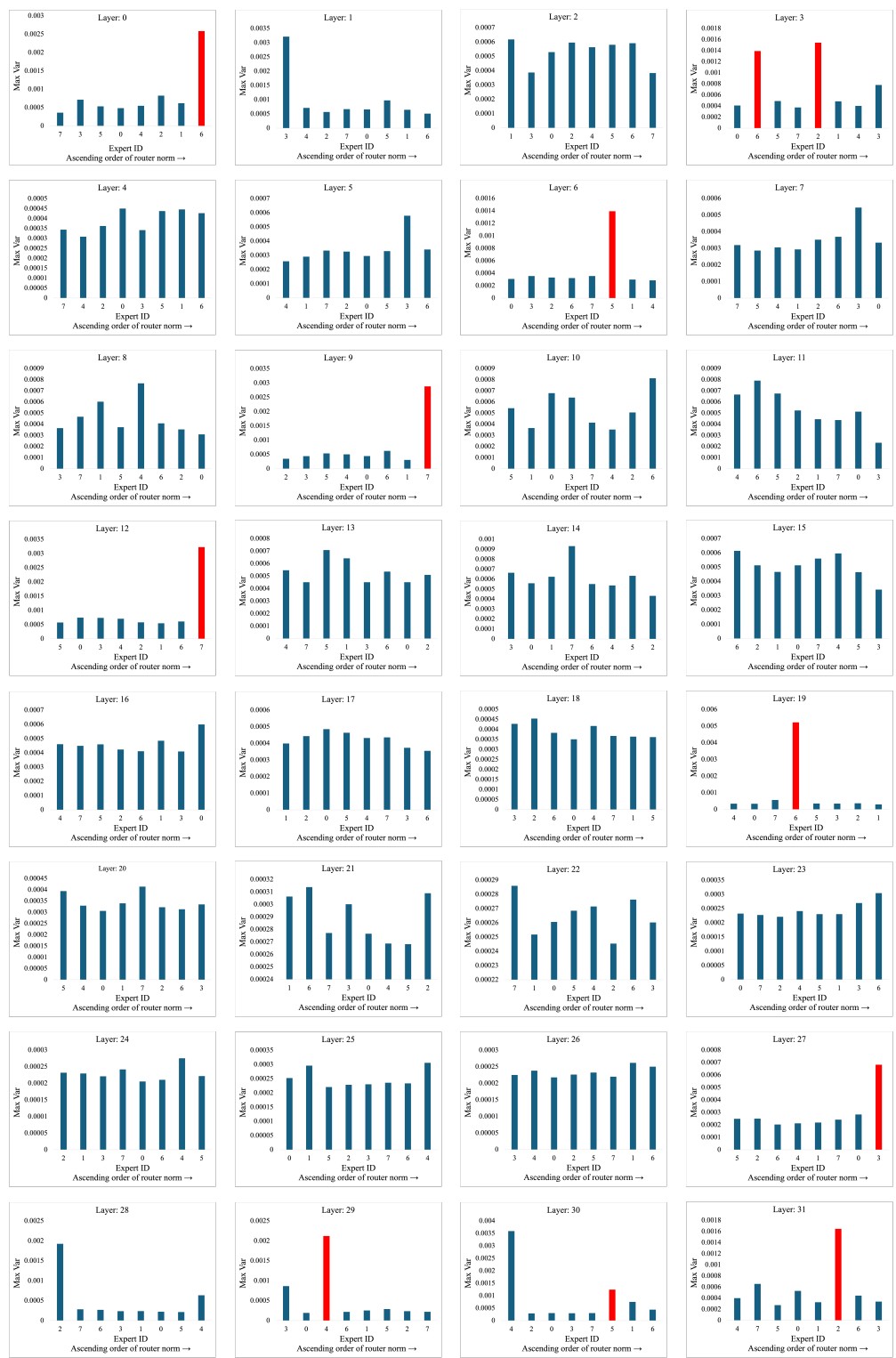

Figure 9: Visualization of the maximum intra-neuron variance of the experts in Mixtral 8x7B. Only a handful of experts (11 out of 256, colored in red) have unusually large MaxVar values.

## F VERIFICATION OF THEORETICAL INSIGHTS IN PRACTICAL MOE MODELS

For accurate verification of our theoretical insights, we first need to identify the task-relevant tokens of different experts, which is hard to determine in practical MoE models. This is because, for some experts, many tokens can be task-relevant, but each of them may appear less frequently in data. On the contrary, for some experts, only a few tokens can be relevant, but each of them may appear very frequently in data. We consider the tokens with high *gating values* as the task-relevant tokens of each expert.

**Larger router norm experts exhibit higher activation.** We verify our claim that larger router norm experts exhibit higher activation in practical MoE models. To verify that claim, we plot the average activations of tokens with top gating values (sampled from the WikiText2 dataset) for each expert in the first five layers of Mixtral 8x7B. The results are provided in Figure 10.

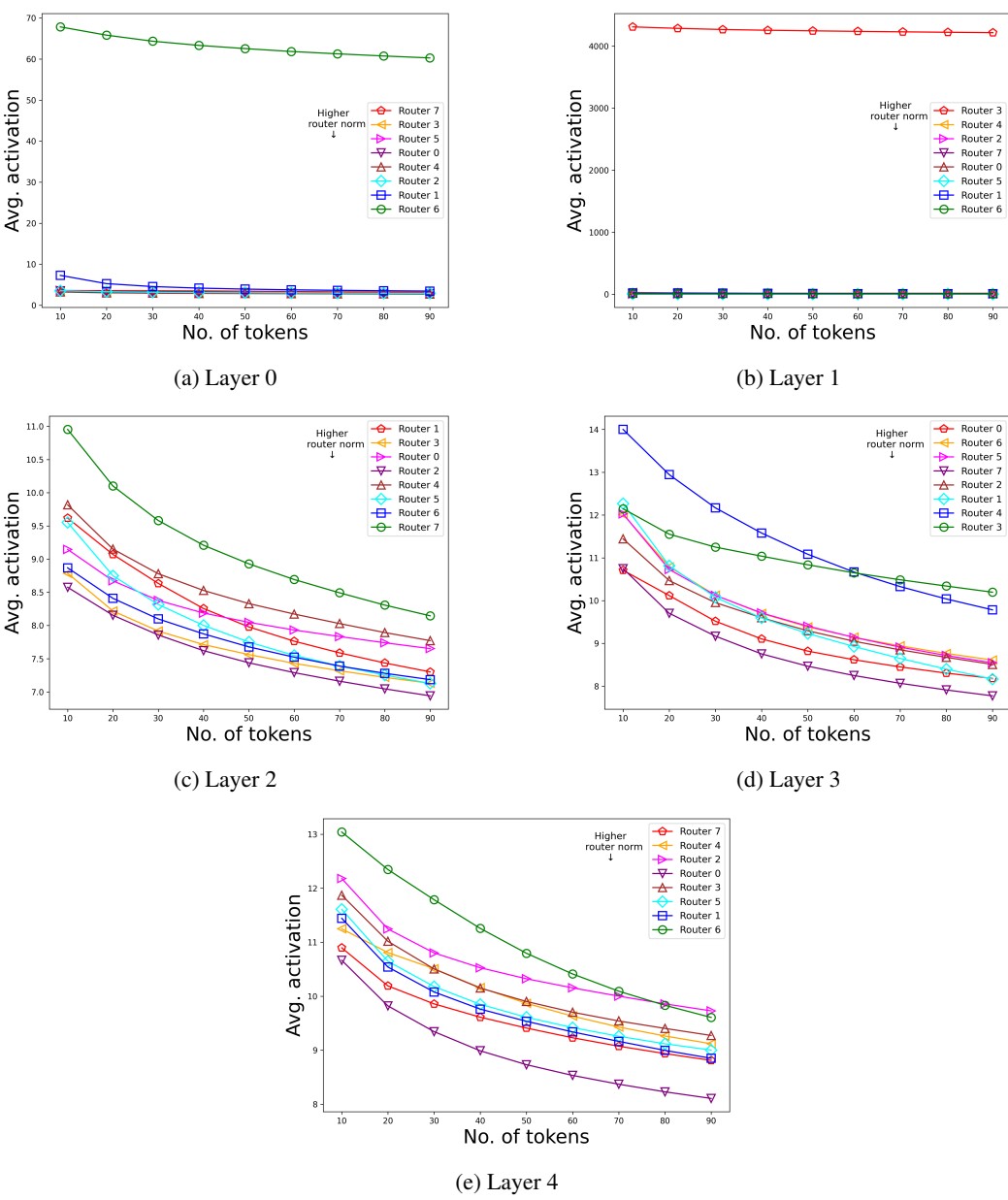

Figure 10: Average activation over tokens with top gating values of different experts of Mixtral 8x7B. The top one/two router norm experts exhibit larger activations.

As we can see, the largest router norm expert (in some cases, the largest two) always exhibits significantly large average activation compared to other experts, except for the second layer. In this case, the lowest one has unusually high activation. However, from our maximum intra-neuron variance, i.e., $\mathrm{MaxVar}$ visualization given in Appendix E, we can see that this expert has an unusually large $\mathrm{MaxVar}$ value than other experts. Therefore, this expert will be placed on higher bit regardless of its position in the router norm order according to our method.

Finally, we visualize the top gating value tokens (sampled from the WikiText2 dataset) through their corresponding model input embeddings for the first MoE block (closest to the model input embeddings) of Mixtral 8x7B. The visualization of the first few tokens with top gating values (highlighted in yellow) of the smallest, and the first two largest router norm experts are provided in Figure 11, along with their adjacent tokens.

. The trackways , called batrachichni , are usually found in strata deposited around
opholis is now considered a prolacertiform reptile . <0x0A><0x0A><0x0A> Later
<0x0A><0x0A> Chad Benekos ( Chizad ) — guitar ( 1994 –
= = <0x0A><0x0A><0x0A><0x0A><0x0A> Oxaziridine derivatives were first reported in the mid
, Malta , Mauritania , Morocco , Oman , Portugal , San Marino ,
<0x0A><0x0A><0x0A><0x0A><0x0A> = = Amalgamation = = <0x0A><0x0A><0x0A><0x0A><0x0A> On
Battalion near Yongsan . Stragglers from this position continued to stream in the next few
omes and Geer recruited guitarist Chizad , bassist Mawk , drummer B.
was to occupy French North Africa : Morocco , Algeria , and Tunisia .
the Zrinski Battalion was amalgamated with several other HV special forces units into the
C Company , but Jenson collected stragglers from it and seized high ground astride this main
<0x0A><0x0A><0x0A><0x0A><0x0A><0x0A><0x0A> = Oxaziridine = <0x0A><0x0A><0x0A><0x0A><0x0A> An
@-@ twenty French aircraft based in Morocco attacked Gibraltar . " On the same day
7 @,@ 740 hectares ( 577 @.@ 4
, 2005 , American Society of Magazine Editors unveiled its list of the top
o Comunitario de San Lorenzo Tenochtitlán near Texistepec . It stands

(a) Expert 7 (smallest router norm)

<0x0A><0x0A><0x0A> The battles of the American Civil War and at Lissa were very influential on the designs
urbed weather associated with a broad low @-@ pressure area south
undred Days , his defeat at the Battle of Waterloo , the pursuit of his army and himself ,
. <0x0A><0x0A><0x0A><0x0A><0x0A> = = Soundtrack = = <0x0A><0x0A><0x0A><0x0A><0x0A> Selected
services used to be provided by the Metropolitan Waterworks and Sewerage System , which served
main guns . <0x0A><0x0A><0x0A> By World War II , the guns used Type 91 armour
1941 , with Operation Barbarossa looming , Felix was amended to Operation
12 August . As it passed through Waterloo Place , on the edge of the Bogside
-@ raids on Great Britain during World War II and Eliot 's declining health . The
German invasion of the USSR , Operation Barbarossa began , and the USSR became an al
0>دودلس اتوxDA><0xA0> Kota Saludong " ( The Kingdom of
of catching enemy commerce raiders , HMS Warspite , which was completed in 188
<0x0A><0x0A><0x0A><0x0A><0x0A> = = = Fire control and sensors = = = <0x0A><0x0A><0x0A><0x0A>
uiapo is referred as the " Old Downtown " where tiangges , markets , bot
<0x0A> Independiente was certified platinum by the Argentine Chamber of Phonograms and Videogram
religious community was dispersed during the English Civil War between Parliamentarians and Royalists but reformed , ending

(b) Expert 1 (second largest router norm)

hill occupied by D Company was in reality the western tip of a large mountain mass that
original estimates were 5 @-@ 10 hPa
as a " progression " as the Doctor was in " a different phase of his life now " and
phoons was 66 % greater than normal . The Philippine
character development , although he felt that it was blunted by the knowledge that the series
Lolita . Nevertheless , Hentzi believed that the film 's themes of materialism and conform
17 . Charles E. Wagner of the Exhibitor 's Trade Review
legs into the ringpost , as he did to Spanky and Gowen , and inter
" You 've got my belt " . Due to a knee injury to Mir , the title unification
years later , the former French colony made its début at the 1964 Summer Olympics in Tokyo
video received a premier on MTV 's Headbangers Ball on June
) was a Jewish sage who lived in the time of the Mishna a prominent supp
graded it to Tropical Storm Parma on October 2
de Park area . The commission was established to arbitrate claims emanating from British seiz
, affecting 98 % of the population . Annually ,
surrounding them that are most likely mucous coatings . They

(c) Expert 6 (largest router norm)

Figure 11: Token visualization of the smallest and the largest two router norm experts

As we can see, the lowest router norm expert (Expert-7) activates on subwords of unusual names/nouns (e.g., batrachich**ni**, prola**certi**form, Chiz**ad**, Oxaz**iri**dine, Moroc**co**, Amalgam**ation**, Stragg**lers**, hect**ares**, Tenoch**titl**án). Each of them is rare in data, but can be critical in the context. This verifies our intuition that the lower router norm experts learn critical but infrequent tokens. On the other hand, the largest router norm expert (Expert-6) activates on many common full words of the English language, such as pronouns (e.g., **that**), prepositions (e.g., **to**, **in**, **on**), etc. The second largest router norm expert (Expert-1) activates on sentences implying war or military operations, which are common in many documents. This verifies our claim that the larger router norm experts learn *more frequent* tokens.

## G  THEORETICAL JUSTIFICATION FOR USING FINAL ROUTER NORM AS A SURROGATE FOR CHANGE IN NORM OF PRETRAINED MoE MODELS

As stated in section 3.3, for the experiments on zero-shot evaluation of pre-trained models, we propose to use the final router norm ($||w_s^{(T)}||$) to approximate the change in the router's norm ($\Lambda_s^{(T)} := ||w_s^{(T)}|| - ||w_0^{(T)}||$), as the randomly initialized model is not publicly available for computing the initial router norm ($||w_s^{(0)}||$). The rationale behind the approximation comes from the fact that the initial routers are generally initialized randomly with small variance (e.g., parameters of DeepSeekMoE are initialized randomly with variance 0.000036 (Dai et al., 2024)). In that case, the initial router norm differences among the routers are too small to alter the change of router norm based order when approximated by final router norm.

Specifically, for any two routers (router 1 and router 2), if router 1's change in norm is larger than router 2's change in norm, i.e., $\Lambda_1^{(T)} > \Lambda_2^{(T)}$, then

$$\Lambda_1^{(T)} - \Lambda_2^{(T)} > 0$$

$$\Rightarrow (||w_1^{(T)}|| - ||w_1^{(0)}||) - (||w_2^{(T)}|| - ||w_2^{(0)}||) > 0$$

$$\Rightarrow (||w_1^{(T)}|| - ||w_2^{(T)}||) - (||w_1^{(0)}|| - ||w_2^{(0)}||) > 0$$

Now, due to the small-variance initialization, $\left|||w_1^{(0)}|| - ||w_2^{(0)}||\right|$ is a very small quantity. Therefore, it is highly likely that $||w_1^{(T)}|| - ||w_2^{(T)}|| > 0$, as long as $\Lambda_1^{(T)} - \Lambda_2^{(T)}$ is not too close to zero.

Based on the above intuition, we provide a formal theorem to justify the claim:

**Theorem G.1.** *Let the routers of the initial model be randomly initialized from $\mathcal{N}(0, \sigma^2)$ with $\sigma = O(1/d)$. Then, with probability at least $1 - \frac{1}{d^2}$, for any two routers $s_1, s_2 \in [k]$ such that $\Lambda_{s_1}^{(T)} - \Lambda_{s_2}^{(T)} = \Omega(1/\sqrt{d})$ we have $||w_{s_1}^{(T)}|| > ||w_{s_2}^{(T)}||$.*

*Proof.* As, $\Lambda_{s_1}^{(T)} - \Lambda_{s_2}^{(T)} = \Omega(1/\sqrt{d})$, we have, $||w_{s_1}^{(T)}|| - ||w_{s_2}^{(T)}|| \geq \left(||w_{s_1}^{(0)}|| - ||w_{s_2}^{(0)}||\right) + \Omega(1/\sqrt{d})$. Now, with probability $1 - \frac{1}{d^2}$, we have $\left|||w_{s_1}^{(0)}|| - ||w_{s_2}^{(0)}||\right| = O(\sigma\sqrt{d}) = O(1/\sqrt{d})$ for our selection of $\sigma$ which completes the proof. $\qquad\square$

The theorem confirms that, for small-variance initialization (i.e., $\sigma = O(1/d)$), the final norm based order preserves the change in norm based order for any two routers unless they are very close to each other (i.e., $\Lambda_{s_1}^{(T)} - \Lambda_{s_2}^{(T)} = O(1/\sqrt{d})$).

# H    PRELIMINARIES

For any fine-tuning iteration $t$, the equation (5) can be represented as,

$$f^{(t)}(x) = \sum_{s=1}^{k} f_s^{(t)}(x) \quad \text{where,} \ f_s^{(t)}(x) = a^{(s)} \sum_{j \in J_s^{(t)}(x)} G_j^{(s,t)} \sum_{r=1}^{m} \text{ReLU}\left( \langle w_r^{(s,t)}, x^{(j)} \rangle \right) \quad (14)$$

Here, $J_s^{(t)}(x) \subset [n]$ is the set of indices of the tokens of the input sequence $x$ that are routed to the expert $s \in [k]$ at time $t$, and $w_r^{(s,t)}$ is the $r$-th column of $W_1^{(s,t)}$. Note that $\left| J_s^{(t)}(x) \right| = l$.

As we analyze the expert-choice routing, for any $j \in J_s^{(t)}(x)$, the gating value $G_j^{(s,t)}$ is evaluated as,

$$G_j^{(s,t)} = \frac{\exp(\langle w_s^{(t)}, x^{(j)} \rangle)}{\sum_{i \in J_s^{(t)}(x)} \exp(\langle w_s^{(t)}, x^{(i)} \rangle)} \quad (15)$$

We analyzed the case where the model is fine-tuned to minimize the Hinge loss

$$\hat{l}^{(t)}(x, y) = \max(1 - yf^{(t)}(x), 0) \quad (16)$$

while the gradients are evaluated on

$$l^{(t)}(x, y) = 1 - yf^{(t)}(x) \quad (17)$$

similar to the setting of Zhang et al. (2023).

For any input $(x, y)$, the gradient for the column $r \in [m]$ of $W_1^{(s,t)}$ is evaluated as,

$$\frac{\partial l^{(t)}(x, y)}{\partial w_r^{(s,t)}} = -ya^{(s)} \sum_{j \in J_s^{(t)}(x)} G_j^{(s,t)} x^{(j)} 1_{\langle w_r^{(s,t)}, x^{(j)} \rangle \geq 0} \quad (18)$$

and the gradient for the router $w_s$ is evaluated as,

$$\frac{\partial l^{(t)}(x, y)}{\partial w_s^{(t)}} = -ya^{(s)} \sum_{j \in J_s^{(t)}(x)} \sigma_j^{(s,t)} G_j^{(s,t)} \sum_{i \in J_s^{(t)}(x) \backslash j} G_i^{(s,t)} (x^{(j)} - x^{(i)}) \quad (19)$$

where, $\sigma_j^{(s,t)} := \sum_{r=1}^{m} \text{ReLU}(\langle w_r^{(s,t)}, x^{(j)} \rangle)$.

We consider that the model is fine-tuned via Stochastic Gradient Descent algorithm (SGD) with batch size $B$, where the expert weights are updated with learning rate $\eta_e$ and the router weights are updated with learning rate $\eta_r$. The batch gradient for the column $r \in [m]$ of $W_1^{(s,t)}$ is evaluated as,

$$\frac{\partial l}{\partial w_r^{(s,t)}} = \frac{1}{B} \sum_{x \in \mathcal{B}_t} \frac{\partial l^{(t)}(x, y)}{\partial w_r^{(s,t)}} \quad (20)$$

and the batch gradient for the router $w_s$ is evaluated as,

$$\frac{\partial l}{\partial w_s^{(t)}} = \frac{1}{B} \sum_{x \in \mathcal{B}_t} \frac{\partial l^{(t)}(x, y)}{\partial w_s^{(t)}} \quad (21)$$

**Notations:**

1. $\tilde{O}(\cdot)$ and $\tilde{\Omega}(\cdot)$ hides the factor $\log(poly(d))$ with a sufficiently large polynomial $poly(\cdot)$

2. With high probability (abbreviated as $w.h.p.$) refers to the probability $1 - \dfrac{1}{poly(d)}$.

**Definitions:**

For any $q \in \mathcal{P} \backslash \{o_1, o_2\}$, we define the activation of the expert $s \in [k]$ by $q$ as,

$\sigma_q^{(s,t)} := \sum_{r=1}^{m} \text{ReLU}(\langle w_r^{(s,t)}, q \rangle).$

For any $v \in \mathcal{P}_r$, we define a complementary expert proficiency measure for the expert $s$ at time $t$ as,

$\bar{p}_v^{(s,t)} := \mathbb{P} \left[ (x,y) \sim \mathcal{D} : \exists j \in J_s^{(t)} \text{ such that } x^{(j)} = v \big| \exists j \in [n] \text{ such that } x^{(j)} = v \right].$

Note that $p_v^{(s,t)} \geq \bar{p}_v^{(s,t)}$.

Without the loss of generality, we assume that for any $s \in S_v$, $\bar{p}_{-v}^{(s,0)} = O(1/d)$.

We define,

- $G_v^{(s,t)}$: Gating value of the token $x^{(j)} = v$ for some $j \in [n]$ and $v \in \mathcal{P}_r$ at expert $s$ and iteration $t$

- $G_q^{(s,t)}$: Gating value of the token $x^{(j)} = q$ for some $j \in [n]$ and $q \in \mathcal{P} \backslash \{o_1, o_2\}$ at expert $s$ and iteration $t$

- $l_q^{(s,t)} := \left| \{ j \in J_s^{(t)}(x) : x^{(j)} = q \} \right|$, is the number of copies of the task-irrelevant vector $q \in \mathcal{P} \backslash \{o_1, o_2\}$ in the set of top $l$ tokens for the input sequence $x$ at expert $s$ and iteration $t$

We define $C_1 := \max \left\{ \left\| w_s^{(0)} \right\| \right\}_{s \in [k]}$, and $C_2 := \max \left\{ \left\| w_r^{(s,0)} \right\| \right\}_{s \in [k], r \in [m]}$.

Without the loss of generality, we analyze the case that $l \geq e^{2C_1}$.

Therefore, $\forall s \in [k], \left\| w_s^{(0)} \right\| \leq \frac{1}{2} \log l$.

We define, $\gamma_v := \frac{|S_v|}{|S_+|}$ for $v \in \{\pm o_1\}$ and $\gamma_v := \frac{|S_v|}{|S_-|}$ for $v \in \{\pm o_2\}$.

Therefore, $\gamma = \max \{\gamma_v\}_{v \in \{o_1, o_2\}}$.

Without the loss of generality, we assume that $\forall v \in \mathcal{P}_r, \gamma_v = \Omega(1)$.

We define, $C_p := \min \{\langle w_s^{(0)}, q - q' \rangle\}_{s \in [k], q \in \mathcal{P} \cup \{-o_1, -o_2\}, q' \in \mathcal{P} \cup \{-o_1, -o_2\} \backslash \{q\}}$.

We assume that $C_p > 0$.

We assume that for any $v \in \mathcal{P}_r$ and any $s \in S_v$, $\frac{\left| \{r \in [m] : \langle w_r^{(s,0)}, v \rangle \geq 0\} \right|}{m} = \Omega(1)$, and $\left| |S_+| - |S_-| \right| = O(\sqrt{k})$.

**Components of the routers' gradients.**

For any input $(x, y) \sim \mathcal{D}$, the router's gradient component of the expert $s \in [k]$ along any task-relevant vector $v \in \mathcal{P}_r$ and along any task-irrelevant vector $q \in \mathcal{P} \backslash \{o_1, o_2\}$ at iteration $t$ are evaluated as follows:

$$\left\langle \frac{\partial l(x,y)}{\partial w_s^{(t)}}, q \right\rangle = \begin{cases} 0 & \text{if } \not\exists j \in J_s^{(t)}(x) \\ & \quad \text{s.t. } x^{(j)} = q \\ ya^{(s)} l_q^{(s,t)} G_q^{(s,t)} \sum_{j \in J_s^{(t)}(x) \backslash \{i: x^{(i)} = q\}} G_j^{(s,t)}(\sigma_j^{(s,t)} - \sigma_q^{(s,t)}) & \text{if } \exists j \in J_s^{(t)}(x) \\ & \quad \text{s.t. } x^{(j)} = q \end{cases}$$

(22)

$$
\left\langle \frac{\partial l(x,y)}{\partial w_s^{(t)}}, v \right\rangle = \begin{cases} 0 & \text{if } \nexists j \in J_s^{(t)}(x) \\ & \text{s.t. } x^{(j)} = v \\ & \text{and } x^{(j)} = -v \\[2ex] ya^{(s)} G_v^{(s,t)}(x) \sum_{j \in J_s^{(t)}(x)/\{i:x^{(i)}=v\}} G_j^{(s,t)} \left( \sigma_j^{(s,t)} - \sigma_v^{(s,t)} \right) & \text{if } \exists j \in J_s^{(t)}(x) \\ & \text{s.t. } x^{(j)} = v \\[2ex] ya^{(s)} G_{-v}^{(s,t)}(x) \sum_{j \in J_s^{(t)}(x)/\{i:x^{(i)}=-v\}} G_j^{(s,t)} \left( \sigma_{-v}^{(s,t)} - \sigma_j^{(s,t)} \right) & \text{if } \exists j \in J_s^{(t)}(x) \\ & \text{s.t. } x^{(j)} = -v \end{cases}
$$
(23)

**Components of the experts' column gradients.**

For any input $(x,y) \sim \mathcal{D}$, the gradient component of the column $r \in [m]$ of $W_1^{(s,t)}$ along any task-relevant vector $v \in \mathcal{P}_r$ and along any task-irrelevant vector $q \in \mathcal{P} \backslash \{o_1, o_2\}$ at iteration $t$ are evaluated as follows:

$$
\left\langle \frac{\partial l(x,y)}{\partial w_r^{(s,t)}}, q \right\rangle = \begin{cases} 0 & \text{if } \langle w_r^{(s,t)}, q \rangle < 0 \\[1ex] 0 & \text{if } \langle w_r^{(s,t)}, q \rangle \geq 0 \text{ but, } \nexists j \in J_s^{(t)}(x) \text{ s.t. } x^{(j)} = q \\[1ex] -ya^{(s)} l_q^{(s,t)} G_q^{(s,t)} & \text{if } \langle w_r^{(s,t)}, q \rangle \geq 0 \text{ and, } \exists j \in J_s^{(t)}(x) \text{ s.t. } x^{(j)} = q \end{cases}
$$
(24)

$$
\left\langle \frac{\partial l(x,y)}{\partial w_r^{(s,t)}}, v \right\rangle = \begin{cases} 0 & \text{if } \langle w_r^{(s,t)}, v \rangle < 0 \text{ but } \nexists j \in J_s^{(t)}(x) \text{ s.t. } x^{(j)} = -v \\[1ex] ya^{(s)} G_{-v}^{(s,t)}(x) & \text{if } \langle w_r^{(s,t)}, v \rangle < 0 \text{ and } \exists j \in J_s^{(t)}(x) \text{ s.t. } x^{(j)} = -v \\[1ex] 0 & \text{if } \langle w_r^{(s,t)}, v \rangle \geq 0 \text{ but } \nexists j \in J_s^{(t)}(x) \text{ s.t. } x^{(j)} = v \\[1ex] -ya^{(s)} G_v^{(s,t)}(x) & \text{if } \langle w_r^{(s,t)}, v \rangle \geq 0 \text{ and } \exists j \in J_s^{(t)}(x) \text{ s.t. } x^{(j)} = v \end{cases}
$$
(25)

# I  PROOF OF LEMMA 4.3

**Proof sketch.** Lemma 4.3 provides the results for training dynamic analysis of the analyzed model. Primarily, our training dynamic analysis provides insights about the learning characteristics of the experts learning different task-relevant tokens. Moreover, the analysis provides necessary bounds of the router norm changes and expert activations required for the mixed-precision quantization analysis, along with the generalization guarantee of the trained model. We categorize the training into two phases:

(i) The expert alignment phase
(ii) The router-expert co-learning phase

**(i) The expert alignment phase.** Given the relative alignments of the routers to different task-relevant tokens, the expert alignment phase confirms that, regardless of the initial alignment of the columns of the expert-weights (i.e., the columns of $W_1^{(s)}$) they sufficiently align with the task-relevant tokens to which their respective routers are initially aligned to. Therefore, the batch gradients during the SGD updates for the router weights maintain large components along the initial alignment direction after this phase. We quantify the number of iterations required to complete this phase of training, along with the bounds of expert activations by different task-relevant tokens after this phase (see Lemma J.5 and Lemma J.6).

**(ii) The router-expert co-learning phase.** After the expert alignment phase, due to the large batch-gradient components of the routers along the initial task-relevant token directions, they become further aligned to these directions in the subsequent updates of SGD. This allows the expert weights to be more aligned with the task-relevant token directions of their respective routers, further increasing the routers' batch-gradient components along these directions. Therefore, the routers and the experts co-learn the task-relevant tokens at least by a quadratic rate. Hence, the model generalizes after this phase of training. However, due to the larger frequency of more-prevalent tokens, the experts learning them receive larger updates in their router and expert weights, allowing larger norm change and expert activations after training, compared to other experts. As shown in Lemma 4.3, we quantify the sufficient number of iterations required to complete the training, along with the router norm changes and expert activation bounds for different experts.

**Lemma I.1** (**Full version of Lemma 4.3**). *Suppose the expert learning rate $\eta_e$, the router learning rate $\eta_r = O\left(\dfrac{\eta_e C_p}{ml^2 C_2^2}\right)$, the batch size $B = \tilde{\Omega}(d^2)$, and the pre-trained model is trained for*

$$T = \Theta\left(\frac{l^2 C_2}{\alpha \eta_e}\sqrt{\frac{\log l}{C_p}}\right) \tag{26}$$

*iterations. Then, the returned $f^{(T)}$ has the following properties:*

*(i) For all $s \in S_v$ and $v \in \mathcal{P}_r = \{\pm o_1, \pm o_2\}$, we have*

$$p_v^{(s,T)} = 1, \quad \bar{p}_{-v}^{(s,T)} = 0, \text{ and}$$

$$\forall x^{(j)} = v \text{ for some } j \in [n], \quad G_j^{(s,T)} > \frac{1}{2}.$$

*(ii) For all $s \in S_{o_i}$ and $s' \in S_{-o_i}$, $i = 1, 2$, we have*

$$\Lambda_{s'}^{(T)} > \Lambda_s^{(T)}.$$

*(iii) For all $s \in S_{o_i}$ and $s' \in S_{-o_i}$, $i = 1, 2$, we have*

$$\sigma_{o_i}^{(s,T)} = \Omega\left(mlC_2\sqrt{\frac{\log l}{C_p}}\right),$$

$$\sigma_{-o_i}^{(s',T)} = \Omega\left(\frac{(1-\alpha)}{\alpha}mlC_2\sqrt{\frac{\log l}{C_p}}\right),$$

$$\frac{\sigma_{-o_i}^{(s',T)}}{\sigma_{o_i}^{(s,T)}} \geq \frac{1-2\alpha}{2\alpha}.$$

*(iv) For all $q \in \mathcal{P} \setminus \{o_1, o_2\}$, $s \in S_v$, $v \in \mathcal{P}_r = \{\pm o_1, \pm o_2\}$, and $v' \in \mathcal{P}_r \setminus \{\pm v\}$, we have*

$$\sigma_q^{(s,T)} = O(mC_2), \quad \sigma_{v'}^{(s,T)} = O(mC_2).$$

*Proof.* (i) Let us consider $s \in S_{o_1}$. From Lemma J.5, we can show that, for $T' = O(\frac{lC_2}{\alpha\eta_e})$, $\forall 0 \leq t \leq T'$, and for any $q \in \mathcal{P}\setminus\{o_1, o_2\}$, $\langle w_s^{(t)}, o_1 - q\rangle < \log l$.

Therefore, using Lemma J.4 and Lemma J.5, by selecting $B = \tilde{\Omega}(d^2)$ we have,
$\langle\dfrac{\partial l}{\partial w_s^{(T')}}, o_1\rangle \leq -\Omega(\dfrac{\alpha m C_2}{l})$. Therefore, $\langle\dfrac{\partial l}{\partial w_s^{(T')}}, -o_1\rangle \geq \Omega(\dfrac{\alpha m C_2}{l})$.

On the other hand, from Lemma J.5, $\left|\langle\dfrac{\partial l}{\partial w_s^{(T')}}, q\rangle\right| = O(\dfrac{m C_2}{d})$.

Therefore, $p_{o_1}^{(s,T'+1)} \geq p_{o_1}^{(s,T')}$, and $\bar{p}_{-o_1}^{(s,T'+1)} \leq \bar{p}_{-o_1}^{(s,T')}$.

Again, as $\langle \frac{\partial l}{\partial w_s^{(T')}}, o_1 \rangle \geq -O(mC_2)$, for our selection of $\eta_r$, we have $\langle w_s^{(T'+1)}, o_1 - q \rangle \leq 2 \log l$.

Therefore,

$\langle \frac{\partial l}{\partial w_s^{(T'+1)}}, o_1 \rangle \leq -\Omega(\frac{\alpha^2 m \eta_e}{l^2}) - \Omega(\frac{\alpha m C_2}{l})$, and hence $\langle \frac{\partial l}{\partial w_s^{(T'+1)}}, -o_1 \rangle \geq \Omega(\frac{\alpha^2 m \eta_e}{l^2}) + \Omega(\frac{\alpha m C_2}{l})$.

On the other hand,

$\langle \frac{\partial l}{\partial w_s^{(T'+1)}}, q \rangle \leq O(\frac{m \eta_e}{d}) + O(\frac{m C_2}{d})$, and $\langle \frac{\partial l}{\partial w_s^{(T'+1)}}, q \rangle \geq -O(\frac{m \eta_e}{d^2}) - O(\frac{m C_2}{d})$.

Therefore, for any $t$ s.t. $\forall T' \leq t' \leq t - 1$, if for all $q \in \mathcal{P} \backslash \{o_1, o_2\}$ it holds that $\langle w_s^{(t')}, o_1 - q \rangle \leq 2 \log l$, by induction we can show that, $p_{o_1}^{(s,T)} \geq p_{o_1}^{(s,t')}$, and $\bar{p}_{-o_1}^{(s,T)} \leq \bar{p}_{-o_1}^{(s,t')}$.

In that case, we have,

$\langle \frac{\partial l}{\partial w_s^{(t)}}, o_1 \rangle \leq -\Omega(\frac{\alpha^2 m \eta_e}{l^2} t) - \Omega(\frac{\alpha m C_2}{l})$, and hence $\langle \frac{\partial l}{\partial w_s^{(t)}}, -o_1 \rangle \geq \Omega(\frac{\alpha^2 m \eta_e}{l^2} t) + \Omega(\frac{\alpha m C_2}{l})$.

On the other hand,

$\langle \frac{\partial l}{\partial w_s^{(t)}}, q \rangle \leq O(\frac{m \eta_e}{d} t) + O(\frac{m C_2}{d})$, and $\langle \frac{\partial l}{\partial w_s^{(t)}}, q \rangle \geq -O(\frac{m \eta_e}{d^2} t) - O(\frac{m C_2}{d})$.

Therefore, $\langle w_s^{(t)}, o_1 - q \rangle \geq \langle w_s^{(T')}, o_1 - q \rangle + \Omega(\frac{\alpha^2 m \eta_e}{l^2} \eta_r (t - T')^2) + \Omega(\frac{\alpha m C_2}{l} \eta_r (t - T'))$.

Now, we can show that, $\langle w_s^{(T')} - w_s^{(0)}, q - o_1 \rangle \leq O(C_p)$. Also, $\left| \langle w_s^{(0)}, o_1 - q \rangle \right| \leq \frac{1}{\sqrt{2}} \log l$.

Therefore, we need $T = O(\frac{l^2 C_2}{\alpha \eta_e} \sqrt{\frac{\log l}{C_p}})$ steps to ensure that, for all task-irrelevant pattern $q$,

$\langle w_s^{(T)}, o_1 - q \rangle > \log l$. In that case, for any $t \geq T'$, $p_{o_1}^{(s,t)} = 1$ and $\forall x^{(j)} = o_1$, $G_j^{(s,t)} \geq \frac{1}{2}$.

Now, if there exists a $q' \in \mathcal{P} \backslash \{o_1, o_2\}$ s.t., $\langle w_s^{(T-1)}, o_1 - q' \rangle > 2 \log l$, then for any $q \in \mathcal{P} \backslash \{o_1, o_2\}$ for which $\langle w_s^{(T-1)}, o_1 - q \rangle \leq \log l$, we have,

$\langle w_s^{(T-1)}, o_1 - q' \rangle = \langle w_s^{(T-1)}, o_1 - q \rangle + \langle w_s^{(T-1)}, q - q' \rangle \leq (1 + \frac{1}{\sqrt{2}}) \log l + O(\frac{l^2}{d} \log l)$ as,

$\langle w_s^{(T-1)}, q - q' \rangle \leq \frac{1}{\sqrt{2}} \log l + O(\frac{l^2}{d} \log l)$. This creates contradiction.

Therefore, $\forall T' \leq t' \leq T$, we have for all task-irrelevant pattern $q$, $\langle w_s^{(t')}, o_1 - q \rangle \leq 2 \log l$.

Now, $\langle w_s^{(T)}, o_1 \rangle > \frac{3}{2} \log l$. Therefore, $\langle w_s^{(T)}, -o_1 \rangle < -\frac{3}{2} \log l$.

Therefore, for any $q \in \mathcal{P} \backslash \{o_1, o_2\}$, $\langle w_s^{(T')}, -o_1 - q \rangle < -\frac{3}{2} \log l - \langle w_s^{(T)}, q \rangle$.

On the other hand, $\left| \langle w_s^{(T)} - w_s^{(0)}, q \rangle \right| = O(\frac{l^2 \log l}{\alpha^2 d})$. Therefore, $\langle w_s^{(T)}, o_3 - q \rangle < 0$, which implies $\bar{p}_{-o_1}^{(s,T)} = 0$.

Similarly, for any $v \in \mathcal{P}_r \backslash \{o_1\}$, and any $s \in S_v$, we can show that $p_v^{(s,T)} = 1$, $\bar{p}_{-v}^{(s,T)} = 0$, and $\forall x^{(j)} = v$ for some $j \in [n]$, $G_j^{(s,T)} \geq \frac{1}{2}$.

(ii) Let $s \in S_{o_1}$ and $s' \in S_{-o_1}$. From the proof of statement (i), we know that, we have for any $q, q' \in \mathcal{P} \backslash \{o_1, o_2\}$ such that, for any $t$ s.t. $t \leq T$, $\left| \langle w_s^{(t)}, q' - q \rangle - \langle w_s^{(0)}, q' - q \rangle \right| = O(\frac{l^2}{d} \log l)$.

Similarly, $\left| \langle w_{s'}^{(t)}, q' - q \rangle - \langle w_{s'}^{(0)}, q' - q \rangle \right| = O(\frac{l^2}{\alpha^2 d} \log l)$.

Now, for any $t$, for any task-irrelevant pattern $q$,
$\langle w_s^{(t+1)}, o_1 - q \rangle \leq \langle w_s^{(0)}, o_1 - q \rangle + O(\alpha m C_2 \eta_r t) + O(\alpha^2 m \eta_e \eta_r t^2)$.

Therefore, at least up to $t = O(T/l)$ iteration, for all $q \in \mathcal{P} \backslash \{o_1, o_2\}$, $\langle w_{s_1}^{(t)}, o_1 - q \rangle \leq 3 \log l$, which implies $\forall t > T_1 = \Omega(T/l)$, for all $q \in \mathcal{P} \backslash \{o_1, o_2\}$, $\langle w_s^{(t)}, o_1 - q \rangle > 3 \log l$, and hence, $\forall x^{(j)} = o_1$,
$G_j^{(s,t)} (1 - G_j^{(s,t)}) \leq \frac{1}{l^2}$.

Therefore, for any $t > T_1$, for any task-irrelevant pattern $q$,
$\langle w_s^{(t+1)}, o_1 \rangle \leq \langle w_s^{(T_1)}, o_1 \rangle + O(\frac{\alpha}{l^2} m C_2 \eta_r (t - T_1)) + O(\frac{\alpha^2}{l^2} m \eta_e \eta_r (t - T_1)^2)$ which implies,
for all task-irrelevant pattern $q$, $\langle w_s^{(T)}, o_1 \rangle \leq \langle w_s^{(T_1)}, o_1 \rangle + O(\log l)$.

Now, as there exists a task-irrelevant pattern $q$ such that, $\langle w_s^{(T_1)}, o_1 - q \rangle \leq 3 \log l$, we have, $\langle w_s^{(T)}, o_1 \rangle - \langle w_s^{(0)}, o_1 \rangle < 4 \log l$.

Now, for any $t$, we have,
$\left| \langle \frac{\partial l}{\partial w_s^{(t)}}, o_2 \rangle \right| \leq O(\frac{m \eta_e}{d} t) + O(m \eta_e) + O(m C_2)$.

Therefore, $\left| \langle w_s^{(T)} - w_s^{(0)}, o_2 \rangle \right| \leq O(\sqrt{C_p})$. Similarly, $\left| \langle w_{s'}^{(T)} - w_{s'}^{(0)}, o_2 \rangle \right| \leq O(\sqrt{C_p})$.

On the other hand, as shown in the proof of (i), for any $q \in \mathcal{P} \backslash \{o_1, o_2\}$, $\left| \langle w_s^{(T)}, q \rangle - \langle w_s^{(0)}, q \rangle \right| = O(\frac{l^2 \log l}{\alpha^2 d})$. Therefore, $\Lambda_s^{(T)} < 4 \log l$.

Now, if $\Lambda_{s'}^{(T)} > \Lambda_s^{(T)}$ does not hold, then $\left\| w_{s'}^{(T)} \right\| < 4.5 \log l$.

Therefore, $\langle w_{s'}^{(T)}, -o_1 \rangle < 4.5 \log l$ which implies, for any $q \in \mathcal{P} \backslash \{o_1, o_2\}$, $\langle w_{s'}^{(T)}, -o_1 - q \rangle < 5 \log l$ as, $\left| \langle w_{s'}^{(T)}, q \rangle - \langle w_{s'}^{(0)}, q \rangle \right| = O(\frac{l^2}{\alpha^2 d} \log l)$.

However, if for all $q \in \mathcal{P} \backslash \{o_1, o_2\}$, $\langle w_{s'}^{(T)}, -o_1 - q \rangle < 5 \log l$, then $\forall x^{(j)} = -o_1$, and $\forall t \leq T, (1 - G_j^{(s',t)}) G_j^{(s',t)} \geq \frac{1}{3l^4}$.

Now, using the same procedure as in the proof of (i), after $T'' \leq \frac{\alpha T}{1 - \alpha}$ steps, we have for all $q \in \mathcal{P} \backslash \{o_1, o_2\}$, $\langle w_{s'}^{(T'')}, -o_1 \rangle > \frac{3}{2} \log l$, which implies, $\forall T'' \leq t \leq T$,
$\langle w_{s'}^{(t+1)}, -o_1 \rangle \geq \frac{3}{2} \log l + \Omega(\frac{(1 - \alpha)}{l^4} m C_2 \eta_r (t - T'')) + \Omega(\frac{(1 - \alpha)^2}{l^4} m \eta_e \eta_r (t - T'')^2)$.

Therefore, $\langle w_{s'}^{(T)}, -o_1 \rangle \geq \frac{3}{2} \log l + \Omega(\frac{(1 - \alpha)^2}{\alpha^2 l^2} \log l)$ which implies $\Lambda_{s'}^{(T)} > \Lambda_s^{(T)}$.

(iii) Let us assume $s \in S_{o_1}$ and $s' \in S_{-o_1}$. Then, $\forall r \in [m]$ such that $\langle w_r^{(s,0)}, o_1 \rangle \geq 0$, from the proof of (i) we have for any $t$, $p_{o_1}^{(s,t)} \geq p_{o_1}^{(s_1,0)}$ which implies, $\forall t \leq T$, $\langle \frac{\partial l}{\partial w_r^{(s,t)}}, o_1 \rangle \leq -\Omega(\frac{\alpha}{l}) + \tilde{O}(\frac{1}{l\sqrt{B}})$ which implies, $\forall r \in [m]$ such that $\langle w_r^{(s,0)}, o_1 \rangle \geq 0$, $\forall t \leq T$,
$\langle w_r^{(s,t+1)}, o_1 \rangle \geq \langle w_r^{(s,t)}, o_1 \rangle + \Omega(\frac{\alpha \eta_e}{l})$ for the choice of $B = \tilde{\Omega}(d^2)$.

Therefore, $\forall r \in [m]$ such that $\langle w_r^{(s,0)}, o_1 \rangle \geq 0$,

$$\langle w_r^{(s,T)}, o_1 \rangle \geq \langle w_r^{(s,0)}, o_1 \rangle + \Omega(\frac{\alpha \eta_e}{l})T = \Omega(lC_2\sqrt{\frac{\log l}{C_p}}), \text{ which implies } \sigma_{o_1}^{(s,T)} = \Omega(mlC_2\sqrt{\log l/C_p}).$$

Again, using the same procedure as in the proof of (i), after $T'' \leq \frac{\alpha T}{1-\alpha}$, we have, $\forall x^{(j)} = -o_1$, $G_j^{(s',T'')} > \frac{1}{2}$ and $\forall r \in [m]$ such that $\langle w_r^{(s',0)}, -o_1 \rangle \geq 0$, we have $\langle w_r^{(s',T'')}, -o_1 \rangle = \Omega(lC_2\sqrt{\frac{\log l}{C_p}})$.

Therefore, we have, $\forall r \in [m]$ such that $\langle w_r^{(s',0)}, -o_1 \rangle \geq 0$,

$$\langle w_r^{(s',T)}, -o_1 \rangle = \Omega\left(\frac{(1-\alpha)}{\alpha}l^2 C_2 \sqrt{\frac{\log l}{C_p}}\right), \text{ which implies } \sigma_{-o_1}^{(s',T)} = \Omega\left(\frac{1-\alpha}{\alpha}ml^2 C_2 \sqrt{\frac{\log l}{C_p}}\right).$$

Similarly, for $s \in S_{o_2}$ and $s' \in S_{-o_2}$, we can show that $\sigma_{o_2}^{(s,T)} = \Omega(mlC_2\sqrt{\log l/C_p})$ and $\sigma_{-o_2}^{(s',T)} = \Omega\left(\frac{1-\alpha}{\alpha}ml^2 C_2 \sqrt{\frac{\log l}{C_p}}\right).$

Now, suppose, $T = K\frac{l^2 C_2}{\alpha \eta_e}\sqrt{\frac{\log l}{C_p}}$, where $K$ is the constant satisfies equation (26).

Then, for any $r \in [m]$ of $s \in S_{o_1}$ such that $\langle w_r^{(s,0)}, o_1 \rangle \geq 0$, we have $\langle w_r^{(s,T)}, o_1 \rangle \leq C_2 + \frac{K}{2}l^2 C_2 \sqrt{\frac{\log l}{C_p}}$.

Again, for any $r \in [m]$ of $s \in S_{o_1}$ such that $\langle w_r^{(s,0)}, o_1 \rangle < 0$, we have $\langle w_r^{(s,T)}, o_1 \rangle < 0$.

Similarly, for any $r \in [m]$ of $s' \in S_{-o_1}$ s.t. $\langle w_r^{(s',0)}, -o_1 \rangle \geq 0$, we have

$$\langle w_r^{(s',T)}, -o_1 \rangle \geq \Omega(lC_2\sqrt{\frac{\log l}{C_p}}) + \frac{K}{2}l^2 C_2 \sqrt{\frac{\log l}{C_p}}. \text{ Therefore, } \sigma_{-o_1}^{(s',T)}/\sigma_{o_1}^{(s,T)} \geq (1-2\alpha)/2\alpha.$$

Similarly, we can show that for any $s \in S_{o_2}$ and $s' \in S_{-o_2}$, $\sigma_{-o_2}^{(s',T)}/\sigma_{o_2}^{(s,T)} \geq (1-2\alpha)/2\alpha$.

(iv) Now, $\forall s \in [k]$, $\forall q \in \mathcal{P}\backslash\{o_1, o_2\}$ and $\forall r \in [m]$ such that $\langle w_r^{(s,0)}, q \rangle \geq 0$, $\forall t$, $\langle \frac{\partial l}{\partial w_r^{(s,t)}}, q \rangle \geq -O(\frac{1}{d}) - \tilde{O}(\frac{1}{\sqrt{B}})$.

Therefore, $\langle w_r^{(s,T')}, q \rangle \leq \langle w_r^{(s,0)}, q \rangle + O(\frac{1}{d}\eta_e T') = C_2 + O(\frac{l^2}{\alpha d}\sqrt{\frac{\log l}{C_p}}C_2) = O(C_2)$ which implies $\sigma_q^{(s,T)} = O(mC_2)$.

Again, $\forall s \in S_+$, $\forall t$ and, $\forall r \in [m]$ such that $\langle w_r^{(s,0)}, o_2 \rangle \geq 0$ we have, $\langle \frac{\partial l}{\partial w_r^{(s,t)}}, o_2 \rangle \geq 0$, and for all $r \in [m]$ s.t. $\langle w_r^{(s,0)}, -o_2 \rangle \geq 0$, $\langle \frac{\partial l}{\partial w_r^{(s,t)}}, -o_2 \rangle \geq 0$ which implies $\langle w_r^{(s,T')}, o_2 \rangle \leq C_2$ and $\langle w_r^{(s,T')}, -o_2 \rangle \leq C_2$. Therefore, $\sigma_{o_2}^{(s,T)} = O(mC_2)$ and $\sigma_{-o_2}^{(s,T)} = O(mC_2)$.

Similarly, we can show that $\forall s \in S_-$, $\sigma_{o_1}^{(s,T)} = O(mC_2)$ and $\sigma_{-o_1}^{(s,T)} = O(mC_2)$.

$\square$

## J   LEMMAS USED TO PROVE LEMMA 4.3

**Lemma J.1.** *Let, $S \subset \mathcal{D}$ such that $p := \mathbb{P}\left[(x, y) \sim \mathcal{D} : (x, y) \in S\right]$. Then, w.h.p. over any randomly sampled batch $\mathcal{B}_t$ of size $B$ at the iteration $t$, $\left| \left| \mathcal{B}_t \cap S \right| - Bp \right| = \tilde{O}\left(\sqrt{B}\right)$.*

*Proof.* Let us define a random variable $X$ associated with any sample $(x, y) \sim \mathcal{D}$ such that,
$$X := \begin{cases} 1 & \text{if } (x, y) \in S \\ 0 & \text{if } (x, y) \notin S \end{cases}$$

Therefore, $X \sim \text{Ber}(p)$.

Now, for any randomly sampled batch $\mathcal{B}_t := \{(x_1, y_1), (x_2, y_2), ..., (x_B, y_B)\}$ of size $B$, we can denote the $B$ i.i.d. random variables following the same distribution as $X$ by $X_1, X_2, ..., X_B$ corresponding to the $B$ samples of the batch, respectively.

Therefore, $\left| \mathcal{B}_t \cap S \right| = \sum_{i=1}^{B} X_i$.

Now, $\mathbb{E}\left[ \left| \mathcal{B}_t \cap S \right| \right] = \sum_{i=1}^{B} \mathbb{E}\left[ X_i \right] = Bp$.

Therefore, using the Hoeffding's inequality, $\mathbb{P}\left[ \left| \left| \mathcal{B}_t \cap S \right| - Bp \right| = \tilde{O}\left(\sqrt{B}\right) \right] \geq 1 - \frac{1}{\text{poly}(d)}$ which completes the proof. $\qquad \square$

**Lemma J.2.** *For any expert $s \in S_v$ with $v, v' \in \{o_1, o_2\}$ such that $v \neq v'$, any $q \in \mathcal{P}\backslash\{o_1, o_2\}$, and any $r \in [m]$, w.h.p. over a randomly sampled batch of size $B$ we can ensure that,*

*(i)* $\left| \langle \frac{\partial l}{\partial w_s^{(0)}}, q \rangle \right| \leq O\left(\frac{mC_2}{d}\right) + \tilde{O}\left(\frac{mC_2}{\sqrt{B}}\right)$

*(ii)* $\left| \langle \frac{\partial l}{\partial w_r^{(s,0)}}, q \rangle \right| \leq O\left(\frac{1}{d}\right) + \tilde{O}\left(\frac{1}{\sqrt{B}}\right)$

*(iii)* $\left| \langle \frac{\partial l}{\partial w_s^{(0)}}, v \rangle \right| \leq O\left(\alpha m C_2\right) + O\left(\frac{(1 - \alpha)}{d} m C_2\right) + \tilde{O}\left(\frac{mC_2}{\sqrt{B}}\right)$

*(iv)* $\langle \frac{\partial l}{\partial w_r^{(s,0)}}, v \rangle \leq -\Omega\left(\frac{\alpha}{l}\right) + \tilde{O}\left(\frac{1}{l\sqrt{B}}\right)$ *if* $\langle w_r^{(s,0)}, v \rangle \geq 0$,

$\langle \frac{\partial l}{\partial w_r^{(s,0)}}, v \rangle \leq O\left(\frac{(1 - \alpha)}{d}\right) + \tilde{O}\left(\frac{1}{\sqrt{B}}\right)$ *if* $\langle w_r^{(s,0)}, v \rangle < 0$,

$\langle \frac{\partial l}{\partial w_r^{(s,0)}}, v \rangle \geq -\frac{\alpha}{2} - \tilde{O}\left(\frac{1}{\sqrt{B}}\right)$ *if* $\langle w_r^{(s,0)}, v \rangle \geq 0$, *and*

$\langle \frac{\partial l}{\partial w_r^{(s,0)}}, v \rangle \geq 0$ *if* $\langle w_r^{(s,0)}, v \rangle < 0$

*(v)* $\left| \langle \frac{\partial l}{\partial w_s^{(0)}}, v' \rangle \right| \leq O\left(m C_2\right) + \tilde{O}\left(\frac{mC_2}{\sqrt{B}}\right)$

*(vi)* $\langle \frac{\partial l}{\partial w_r^{(s,0)}}, v' \rangle \geq 0$ *if* $\langle w_r^{(s,0)}, v' \rangle \geq 0$,

$\langle \frac{\partial l}{\partial w_r^{(s,0)}}, v' \rangle \geq -O\left((1 - \alpha)\right) - \tilde{O}\left(\frac{1}{\sqrt{B}}\right)$ *if* $\langle w_r^{(s,0)}, v' \rangle < 0$,

$$\langle \frac{\partial l}{\partial w_r^{(s,0)}}, v' \rangle \leq O(\alpha) + \tilde{O}\left(\frac{1}{\sqrt{B}}\right), \text{ if } \langle w_r^{(s,0)}, v' \rangle \geq 0, \text{ and}$$

$$\langle \frac{\partial l}{\partial w_r^{(s,0)}}, v' \rangle \leq 0 \text{ if } \langle w_r^{(s,0)}, v' \rangle < 0$$

*Proof.* For any $v \in \mathcal{P}_r$ and any $q \in \mathcal{P}\backslash\{o_1, o_2\}$,

$$\left|\sigma_v^{(s,0)} - \sigma_q^{(s,0)}\right| = \left|\sum_{r \in [m]} \text{ReLU}(\langle w_r^{(s,0)}, v \rangle) - \sum_{r \in [m]} \text{ReLU}(\langle w_r^{(s,0)}, q \rangle)\right| = O(mC_2).$$

Similarly, for any $q, q' \in \mathcal{P}\backslash\{o_1, o_2\}$ such that $q \neq q'$, $\left|\sigma_q^{(s,0)} - \sigma_{q'}^{(s,0)}\right| = O(mC_2)$.

We denote $\mathcal{B}_0$ as the randomly sampled batch before the first update of SGD.

**(i)** $\langle \frac{\partial l}{\partial w_s^{(0)}}, q \rangle = \frac{1}{B}\sum_{x \in \mathcal{B}_0}\langle \frac{\partial l(x,y)}{\partial w_s^{(0)}}, q \rangle$

Let us define the set $\bar{S}_q^{J_s^{(0)}} := \{(x,y) \sim \mathcal{D} : \exists j \in J_s^{(0)} \text{ s.t. } x^{(j)} = q\}$ and,

$p_q^{(s,0)} := \mathbb{P}\left[(x,y) \sim \mathcal{D} : (x,y) \in \bar{S}_q^{J_s^{(0)}}\right]$

Here, $p_q^{(s,0)} = O(\frac{1}{d})$

Therefore, $\langle \frac{\partial l}{\partial w_s^{(0)}}, q \rangle = \frac{1}{B}\sum_{x \in \mathcal{B}_0 \cap \bar{S}_q^{J_s^{(0)}}}\langle \frac{\partial l(x,y)}{\partial w_s^{(0)}}, q \rangle + \frac{1}{B}\sum_{x \in \mathcal{B}_0 \cap \mathcal{D}\backslash\bar{S}_q^{J_s^{(0)}}}\langle \frac{\partial l(x,y)}{\partial w_s^{(0)}}, q \rangle$

Now, from equation (22), for any $(x,y) \in \mathcal{D}\backslash\bar{S}_q^{J_s^{(0)}}$, $\langle \frac{\partial l(x,y)}{\partial w_s^{(0)}}, q \rangle = 0$

Therefore, $\langle \frac{\partial l}{\partial w_s^{(0)}}, q \rangle = \frac{1}{B}\sum_{x \in \mathcal{B}_0 \cap \bar{S}_q^{J_s^{(0)}}}\langle \frac{\partial l(x,y)}{\partial w_s^{(0)}}, q \rangle$

Now, for any $(x,y)$, $l_q^{(s,0)} G_q^{(s,0)} \leq 1$

Therefore, as $|\sigma_v^{(s,0)} - \sigma_q^{(s,0)}| = O(mC_2)$ and for any $q'\mathcal{P}\backslash\{o_1, o_2\}$ such that $q \neq q'$, $|\sigma_q^{(s,0)} - \sigma_{q'}^{(s,0)}| = O(mC_2)$, from equation (22), $\left|\langle \frac{\partial l}{\partial w_s^{(0)}}, q \rangle\right| \leq \frac{\left|\mathcal{B}_0 \cap \bar{S}_q^{J_s^{(0)}}\right|}{B}O(mC_2)$

Now, from Lemma J.1, *w.h.p.*, $\frac{\left|\mathcal{B}_0 \cap \bar{S}_q^{J_s^{(0)}}\right|}{B} \leq p_q^{(s,0)} + \tilde{O}\left(\frac{1}{\sqrt{B}}\right)$ which implies,

$\left|\langle \frac{\partial l}{\partial w_s^{(0)}}, q \rangle\right| \leq O\left(\frac{mC_2}{d}\right) + \tilde{O}\left(\frac{mC_2}{\sqrt{B}}\right).$

**(ii)** Using equation (24) and the fact that for any $(x,y) \sim \mathcal{D}$, $l_q^{(s,t)} G_q^{(s,t)} \leq 1$ and by following the same procedure as in the proof of the statement (i) we can complete the proof.

**(iii)** Let us define the set, $\bar{S}_v^{J_s^{(0)}} := \{(x,y) \sim \mathcal{D} : \exists j \in J_s^{(0)} \text{ s.t. } x^{(j)} = v\}$.

Now,

$$\mathbb{P}\left[(x,y) \sim \mathcal{D} : (x,y) \in \bar{S}_v^{J_s^{(0)}}\right]$$

$$= \mathbb{P}\left[(x,y) \sim \mathcal{D} : (x,y) \in \bar{S}_v^{J_s^{(0)}} \,\middle|\, y = +1 \text{ and } \exists j \in [n] \text{ s.t. } x^{(j)} = v\right]$$

$$\times \mathbb{P}\left[(x,y) \sim \mathcal{D} : y = +1 \text{ and } \exists j \in [n] \text{ s.t. } x^{(j)} = v\right]$$

$$\leq \frac{\alpha}{2}$$

On the other hand, $\bar{p}_{-v}^{(s,0)} = O(1/d)$.

Now, using equation (23), by following the same procedure as in the proof of statement (i), we can complete the proof.

**(iv)** Let us define the set, $S_v^{J_s^{(0)}} := \left\{(x,y) \sim \mathcal{D} : \exists j \in J_s^{(0)} \text{ s.t. } x^{(j)} = v \text{ and } G_v^{(s,0)} \geq \frac{1}{l}\right\}$.

Now,

$$\mathbb{P}\left[(x,y) \sim \mathcal{D} : (x,y) \in S_v^{J_s^{(0)}}\right]$$

$$= \mathbb{P}\left[(x,y) \sim \mathcal{D} : (x,y) \in S_v^{J_s^{(0)}} \,\middle|\, y = +1 \text{ and } \exists j \in [n] \text{ s.t. } x^{(j)} = v\right]$$

$$\times \mathbb{P}\left[(x,y) \sim \mathcal{D} : y = +1 \text{ and } \exists j \in [n] \text{ s.t. } x^{(j)} = v\right]$$

$$= p_v^{(s,0)} \frac{\alpha}{2} = \Omega(\alpha) \qquad \left[\text{As, } p_v^{(s,0)} = \Omega(1)\right]$$

On the other hand, $\bar{p}_{-v}^{(s,0)} = O(1/d)$.

Now, using equation (25) and by following the same procedure as in the proof of the statement (i) we can complete the proof.

**(v)** Using equation (23) and by following the same procedure as in the statements (iii) and (i) we can complete the proof.

**(vi)** Using equation (25) and following the same procedure as in the proof of statement (ii) and (iv) we can complete the proof.

$\square$

**Lemma J.3.** *For any expert $s \in S_v$ with $v, v' \in \{-o_1, -o_2\}$ such that $v \neq v'$, any $q \in \mathcal{P}\backslash\{o_1, o_2\}$, and any $r \in [m]$, w.h.p. over a randomly sampled batch of size $B$ we can ensure that,*

*(i)* $\left|\langle \frac{\partial l}{\partial w_s^{(0)}}, q\rangle\right| \leq O\left(\frac{mC_2}{d}\right) + \tilde{O}\left(\frac{mC_2}{\sqrt{B}}\right)$

*(ii)* $\left|\langle \frac{\partial l}{\partial w_r^{(s,0)}}, q\rangle\right| \leq O\left(\frac{1}{d}\right) + \tilde{O}\left(\frac{1}{\sqrt{B}}\right)$

*(iii)* $\left|\langle \frac{\partial l}{\partial w_s^{(0)}}, v\rangle\right| \leq O\left((1-\alpha)mC_2\right) + O\left(\frac{\alpha}{d}mC_2\right) + \tilde{O}\left(\frac{mC_2}{\sqrt{B}}\right)$

*(iv)* $\langle \frac{\partial l}{\partial w_r^{(s,0)}}, v\rangle \leq -\Omega\left(\frac{(1-\alpha)}{l}\right) + \tilde{O}\left(\frac{1}{l\sqrt{B}}\right)$ *if* $\langle w_r^{(s,0)}, v\rangle \geq 0$,

   $\langle \frac{\partial l}{\partial w_r^{(s,0)}}, v\rangle \leq O\left(\frac{\alpha}{d}\right) + \tilde{O}\left(\frac{1}{\sqrt{B}}\right)$ *if* $\langle w_r^{(s,0)}, v\rangle < 0$,

$$\langle \frac{\partial l}{\partial w_r^{(s,0)}}, v \rangle \geq -\frac{(1-\alpha)}{2} - \tilde{O}\left(\frac{1}{\sqrt{B}}\right) \text{ if } \langle w_r^{(s,0)}, v \rangle \geq 0, \text{ and}$$

$$\langle \frac{\partial l}{\partial w_r^{(s,0)}}, v \rangle \geq 0 \text{ if } \langle w_r^{(s,0)}, v \rangle < 0$$

(v) $\left| \langle \frac{\partial l}{\partial w_s^{(0)}}, v' \rangle \right| \leq O\left(mC_2\right) + \tilde{O}\left(\frac{mC_2}{\sqrt{B}}\right)$

(vi) $\langle \frac{\partial l}{\partial w_r^{(s,0)}}, v' \rangle \geq 0 \text{ if } \langle w_r^{(s,0)}, v' \rangle \geq 0,$

$$\langle \frac{\partial l}{\partial w_r^{(s,0)}}, v' \rangle \geq -O\left(\alpha\right) - \tilde{O}\left(\frac{1}{\sqrt{B}}\right) \text{ if } \langle w_r^{(s,0)}, v' \rangle < 0,$$

$$\langle \frac{\partial l}{\partial w_r^{(s,0)}}, v' \rangle \leq O\left((1-\alpha)\right) + \tilde{O}\left(\frac{1}{\sqrt{B}}\right), \text{ if } \langle w_r^{(s,0)}, v' \rangle \geq 0, \text{ and}$$

$$\langle \frac{\partial l}{\partial w_r^{(s,0)}}, v' \rangle \leq 0 \text{ if } \langle w_r^{(s,0)}, v' \rangle < 0$$

*Proof.* Using the same procedure as in Lemma J.2, we can complete the proof. $\square$

**Lemma J.4.** *For any expert $s \in [k]$, any $v \in \mathcal{P}_r$, and at any iteration $t$, if every $q \in \mathcal{P} \setminus \{o_1, o_2\}$ that satisfies the condition $\langle w_s^{(t)}, q \rangle < \langle w_s^{(t)}, v \rangle$ also satisfies the condition $\langle w_s^{(t)}, v \rangle - \langle w_s^{(t)}, q \rangle \leq 2 \log l$, then for any $j \in J_s^{(t)}$ where $x^{(j)} = v$ and $G_j^{(s,t)} \geq 1/l$, we have, $G_j^{(s,t)}(1 - G_j^{(s,t)}) \geq \frac{1}{4l}$*

*Proof.* If for all $q \in \mathcal{P} \setminus \{o_1, o_2\}$ with $\langle w_s^{(t)}, q \rangle < \langle w_s^{(t)}, v \rangle$ we have, $\langle w_s^{(t)}, v \rangle - \langle w_s^{(t)}, q \rangle \leq 2 \log l$, then $\forall(x,y) \sim \mathcal{D}$ s.t. $\exists j \in J_s^{(t)}(x)$ with $x^{(j)} = v$ and $G_j^{(s,t)}(x) \geq G_i^{(s,t)}(x), \forall i \in J_s^{(t)}(x)$ and $i \neq j$ we have, $G_j^{(s,t)}(x)(1 - G_j^{(s,t)}(x)) \geq \min\{\frac{\frac{(l-1)}{l^2}}{(1 + \frac{(l-1)}{l^2})^2}, \frac{(l-1)}{l^2}\} = \frac{\frac{(l-1)}{l^2}}{(1 + \frac{(l-1)}{l^2})^2}.$

Now, $\frac{\frac{(l-1)}{l^2}}{(1 + \frac{(l-1)}{l^2})^2} = \frac{l^2(l-1)}{(l^2 + l - 1)^2}.$

Now, let there exists a constant $C > 0$ such that $\frac{l^2(l-1)}{(l^2 + l - 1)^2} \geq \frac{C}{l} \Leftrightarrow l^4(1-C) - l^3(1 + 2C) + Cl^2 + 2Cl - C \geq 0.$

Now, $Cl^2 + 2Cl - C > 0$ as $l \geq 2$. Therefore, $l^3(1 + 2C) \leq l^4(1 - C)$ satisfies $l^4(1 - C) - l^3(1 + 2C) + Cl^2 + 2Cl - C \geq 0.$

Now, $l^3(1 + 2C) \leq l^4(1 - C) \Leftrightarrow C \leq \frac{l-1}{l+2}$. Now, $\frac{l-1}{l+2} \geq \frac{1}{4}$ as $l \geq 2$. Hence, picking $C = \frac{1}{4}$ satisfies that $\frac{l^2(l-1)}{(l^2 + l - 1)^2} \geq \frac{1}{4l}$ which implies $G_j^{(s,t)}(x)(1 - G_j^{(s,t)}(x)) \geq \frac{1}{4l}$. $\square$

**Lemma J.5.** *For any expert $s \in S_v$ such that $v \in \{o_1, o_2\}$, and $\forall q \in \mathcal{P} \setminus \{o_1, o_2\}$, by selecting $\eta_r = O\left(\frac{\eta_e C_p}{ml^2 C_2^2}\right)$ and $B = \tilde{\Omega}\left(d^2\right)$, we can ensure that after $T' = O\left(\frac{lC_2}{\alpha \eta_e}\right)$ iterations,*

(i) $\sigma_v^{(s,T')} = \Omega\left(mC_2\right), \sigma_{-v}^{(s,T')} = O(mC_2), \sigma_q^{(s,T')} = O(mC_2)$

(ii) $p_v^{(s,T')} \geq p_v^{(s,0)}$ and, $\bar{p}_{-v}^{(s,T')} \leq \bar{p}_{-v}^{(s,0)}$

*Proof.* Suppose $v = o_1$. From the statement (i) of the Lemma J.2, *w.h.p.* over a randomly sampled batch, $\left| \langle \frac{\partial l}{\partial w_s^{(0)}}, q \rangle \right| \leq O(\frac{mC_2}{d}) + \tilde{O}(\frac{mC_2}{\sqrt{B}})$

Therefore, $\left| \langle w_s^{(1)}, q \rangle - \langle w_s^{(0)}, q \rangle \right| \leq O(\frac{mC_2}{d}\eta_r) + \tilde{O}(\frac{mC_2}{\sqrt{B}}\eta_r)$.

On the other hand, from the statement (iii) of the Lemma J.2, *w.h.p.* over a randomly sampled batch,
$$\left| \langle \frac{\partial l}{\partial w_s^{(0)}}, o_1 \rangle \right| \leq O(\alpha mC_2) + O\left( \frac{(1-\alpha)}{d}mC_2 \right) + \tilde{O}(\frac{mC_2}{\sqrt{B}})$$

Therefore, $\left| \langle w_s^{(1)}, o_1 \rangle - \langle w_s^{(0)}, o_1 \rangle \right| \geq O(\alpha mC_2\eta_r) + O\left( \frac{(1-\alpha)}{d}mC_2\eta_r \right) + \tilde{O}(\frac{mC_2}{\sqrt{B}}\eta_r)$.

Now, by selecting $\eta_r = O(\frac{C_p}{\alpha mC_2})$ and $B = \tilde{\Omega}\left( \frac{1}{\alpha^2} \right)$, for $\langle w_s^{(0)}, q \rangle < \langle w_s^{(0)}, o_1 \rangle$ we get ,
$\langle w_s^{(1)}, o_1 \rangle - \langle w_s^{(1)}, q \rangle = \Omega(C_p)$ which ensures that $p_{o_1}^{(s,1)} \geq p_{o_1}^{(s,0)}$.

Similarly, we can show that, $\langle w_s^{(1)}, o_1 - q \rangle \leq 2\log l$ and $\bar{p}_{-o_1}^{(s,1)} \leq \bar{p}_{-o_1}^{(s,0)}$.

Now, for any $r \in [m]$ such that $\langle w_r^{(s,0)}, o_1 \rangle \geq 0$, from the statement (iv) of the Lemma J.2, *w.h.p.* $\langle \frac{\partial l}{\partial w_r^{(s,0)}}, o_1 \rangle \leq -\Omega\left( \frac{\alpha}{l} \right) + \tilde{O}(\frac{1}{l\sqrt{B}}))$, and for any $r \in [m]$ such that $\langle w_r^{(s,0)}, o_1 \rangle < 0$,
$\langle \frac{\partial l}{\partial w_r^{(s,0)}}, o_1 \rangle \leq O(\frac{(1-\alpha)}{d}) + \tilde{O}(1/\sqrt{B})$, which implies, for $\langle w_r^{(s,0)}, o_1 \rangle \geq 0$,
$\langle w_r^{(s,1)}, o_1 \rangle \geq \langle w_r^{(s,0)}, o_1 \rangle + \Omega\left( \frac{\alpha\eta_e}{l} \right) - \tilde{O}(\frac{\eta_e}{l\sqrt{B}})$, and for $\langle w_r^{(s,0)}, o_1 \rangle < 0$, $\langle w_r^{(s,1)}, o_1 \rangle < 0$.
Hence, $\sigma_{o_1}^{(s,1)} \geq \sigma_{o_1}^{(s,0)} + \Omega(\frac{\alpha m\eta_e}{l}) - \tilde{O}(\frac{m\eta_e}{l\sqrt{B}})$.

Similarly, using statement (ii), (iii), and (iv) of Lemma J.2, we can show that,

$$\sigma_{o_1}^{(s,1)} \leq \sigma_{o_1}^{(s,0)} + O(\alpha m\eta_e) + \tilde{O}(\frac{m}{\sqrt{B}}\eta_e),$$

$$\sigma_{-o_1}^{(s,1)} \leq \sigma_{-o_1}^{(s,0)} + O\left( \frac{1-\alpha}{d}m\eta_e \right) + \tilde{O}\left( \frac{1}{\sqrt{B}}m\eta_e \right), \sigma_3^{(s,1)} \geq \sigma_3^{(s,0)},$$

$$\left| \sigma_q^{(s,1)} - \sigma_q^{(s,0)} \right| \leq O(\frac{m}{d}\eta_e) + \tilde{O}(\frac{m\eta_e}{\sqrt{B}}).$$

Therefore, by selecting $B = \tilde{\Omega}(d^2)$ we get,

$$\langle \frac{\partial l}{\partial w_s^{(1)}}, q \rangle \leq O(\frac{m\eta_e}{d}) + O(\frac{mC_2}{d}), \langle \frac{\partial l}{\partial w_s^{(1)}}, q \rangle \geq -O(\frac{m\eta_e}{d^2}) - O(\frac{mC_2}{d}),$$

$$\langle \frac{\partial l}{\partial w_s^{(1)}}, o_1 \rangle \leq O(\alpha mC_2) - \Omega(\frac{\alpha^2 m\eta_e}{l}), \langle \frac{\partial l}{\partial w_s^{(1)}}, o_1 \rangle \geq -O(\alpha mC_2) - O(\alpha^2 m\eta_e),$$

$$\langle \frac{\partial l}{\partial w_s^{(1)}}, -o_1 \rangle \leq O(\alpha mC_2) + O(\alpha^2 m\eta_e), \langle \frac{\partial l}{\partial w_s^{(1)}}, -o_1 \rangle \geq -O(\alpha mC_2) + \Omega(\frac{\alpha^2 m\eta_e}{l}),$$

(Condition 1) Suppose, there exists a $T'$ such that $\forall 0 \leq t \leq T', p_{o_1}^{(s,t)} \geq p_{o_1}^{(s,0)}, \bar{p}_{-o_1}^{(s,t)} \leq \bar{p}_{-o_1}^{(s,0)}$.

Now, if condition 1 holds then, $\sigma_{o_1}^{(s,T)} \geq \sigma_{-o_1}^{(s,0)} + \Omega(\frac{\alpha m \eta_e}{l} T)$, which implies we need $T' = O(\frac{lC_2}{\alpha \eta_e})$ steps to ensure that, $\sigma_{o_1}^{(s,T)} = \Omega(mC_2)$. Also, as $\sigma_{-o_1}^{(s,T)} \leq \sigma_{-o_1}^{(s,0)} + O\left(\frac{1-\alpha}{d} m \eta_e T\right)$, we have, $\sigma_{-o_1}^{(s,T)} = O(mC_2)$. Similarly, $\sigma_q^{(s,T)} = O(mC_2)$.

Again, if condition 1 holds, then using Lemma J.2 and equation (22), and equation (25) we can show that, $\forall 0 \leq t \leq T'$, we have,

$$\langle \frac{\partial l}{\partial w_s^{(t)}}, q \rangle \leq O(\frac{m\eta_e}{d} t) + O(\frac{mC_2}{d}), \langle \frac{\partial l}{\partial w_s^{(t)}}, q \rangle \geq -O(\frac{m\eta_e}{d^2} t) - O(\frac{m\eta_e}{d}) - O(\frac{mC_2}{d}),$$

$$\langle \frac{\partial l}{\partial w_s^{(1)}}, o_1 \rangle \leq O(\alpha m C_2) - \Omega(\frac{\alpha^2 m \eta_e}{l} t), \langle \frac{\partial l}{\partial w_s^{(t)}}, o_1 \rangle \geq -O(\alpha m C_2) - O(\alpha^2 m \eta_e t),$$

$$\langle \frac{\partial l}{\partial w_s^{(t)}}, -o_1 \rangle \leq O(\alpha m C_2) + O(\alpha^2 m \eta_e t), \langle \frac{\partial l}{\partial w_s^{(t)}}, -o_1 \rangle \geq -O(\alpha m C_2) + \Omega(\frac{\alpha^2 m \eta_e}{l} t).$$

Therefore, by selecting $\eta_r = O(\frac{C_p}{\alpha m C_2} \frac{1}{T'}) = O(\frac{C_p \eta_e}{\alpha m l C_2^2})$, we can ensure that, condition 1 holds for our selection of $T'$.

Similarly, we can prove the case of $v = o_2$. $\qquad \square$

**Lemma J.6.** *For any expert $s \in S_v$ such that $v \in \{-o_1, -o_2\}$, and $\forall q \in \mathcal{P} \backslash \{o_1, o_2\}$, by selecting $\eta_r = O\left(\frac{\eta_e C_p}{m l^2 C_2^2}\right)$ and $B = \tilde{\Omega}\left(d^2\right)$, we can ensure that after $T' = O\left(\frac{lC_2}{(1-\alpha)\eta_e}\right)$ iterations,*

*(i) $\sigma_v^{(s,T')} = \Omega\left(mC_2\right), \sigma_{-v}^{(s,T')} = O(mC_2), \sigma_q^{(s,T')} = O(mC_2)$*

*(ii) $p_v^{(s,T')} \geq p_v^{(s,0)}$ and, $\bar{p}_{-v}^{(s,T')} \leq \bar{p}_{-v}^{(s,0)}$*

*Proof.* The proof is similar to the proof of Lemma J.5. $\qquad \square$

## K  PROOF OF THEOREM 4.4

**Proof sketch.** The results of Theorem 4.4 are provided by the post-training quantization analysis. Given the experts' activation bounds of the trained model, we estimate how much the activations produced by the quantized weights are allowed to deviate from their original values yet correctly classify the sequences. As the activations of the experts that learned more prevalent tokens are larger compared to the experts that learned less prevalent tokens, the former are allowed to deviate more than the latter. We use the maximum allowable deviations of expert activations for the two groups of experts (i.e., the experts that learned less prevalent tokens, and the experts that learned more prevalent tokens) to estimate corresponding quantization bin sizes via equation (2). Finally, we evaluate the sufficient bit-widths of the two groups of experts from their corresponding maximum allowable bin sizes.

**Theorem K.1** (**Full version of Theorem 4.4**). *Suppose the number of fine-tuning iterations satisfies $T = \Theta(\frac{l^2 C_2}{\alpha \eta_e} \sqrt{\frac{\log l}{C_p}})$, and $\max_{r \in [m]} Var_r^{(s,T)} = \Theta(1)$ for every expert $s$. If $\kappa \geq \gamma$, and the two quantization levels satisfy*

$$b_h \geq \log_2(1 + \Omega(d\sqrt{C_p \log(kmd^2)/l^2 C_2^2 \log l})) \tag{27}$$

*and*

$$b_l \geq \log_2(1 + \frac{\alpha}{1-\alpha} \Omega(d\sqrt{C_p \log(kmd^2)/l^2 C_2^2 \log l})), \tag{28}$$

*then w.h.p. the quantized model has guaranteed generalization, i.e.,*

$$\mathbb{P}[\forall(x,y) \sim \mathcal{D} : y f_Q^{(T)}(x) > 0] = 1. \tag{29}$$

*Proof.* For any $r \in [m]$ of $s \in [k]$, we denote the quantized representation of $w_r^{(s,T)} = \left[ w_{r_1}^{(s,T)}, w_{r_2}^{(s,T)}, ..., w_{r_d}^{(s,T)} \right]^T$ by, $w_r^{(s,T;Q)} = \left[ w_{r_1}^{(s,T;Q)}, w_{r_2}^{(s,T;Q)}, ..., w_{r_d}^{(s,T;Q)} \right]^T$ $= \left[ w_{r_1}^{(s,T)} + \Delta w_{r_1}^{(s,T;Q)}, w_{r_2}^{(s,T)} + \Delta w_{r_2}^{(s,T;Q)}, ..., w_{r_d}^{(s,T)} + \Delta w_{r_d}^{(s,T;Q)} \right]^T$.

Here, for any $i \in [d], \Delta w_{r_i}^{(s,T;Q)}$ is the quantization-noise generated from the quantization of the weight $w_{r_i}^{(s,T)}$.

Now, over the randomness of the pre-trained model, for any $r \in [m]$ of any $s \in [k]$, for any $i \in [d], \Delta w_{r_i}^{(s,T;Q)} \sim \text{Unif} \left[ -\frac{\Delta_r^{(s)}}{2}, \frac{\Delta_r^{(s)}}{2} \right]$, where $\Delta_r^{(s)}$ is the quantization bin size of the column $r \in [m]$ of the expert $s \in [k]$. Here we assume that, for $i_1, i_2 \in [d]$ s.t. $i_1 \neq i_2, \Delta w_{r_{i_1}}^{(s,T;Q)}$ and $\Delta w_{r_{i_2}}^{(s,T;Q)}$ are independent to each other. Similarly, we assume that for any $r_1, r_2 \in [m]$ s.t. $r_1 \neq r_2$, $\Delta w_{r_{1_{i_1}}}^{(s,T;Q)}, \Delta w_{r_{2_{i_2}}}^{(s,T;Q)}, \Delta w_{r_{1_{i_2}}}^{(s,T;Q)}$ and, $\Delta w_{r_{2_{i_1}}}^{(s,T;Q)}$, are independent to each other. We further assume that, for any $s_1, s_2 \in [k]$ s.t. $s_1 \neq s_2, \Delta w_{r_{1_{i_1}}}^{(s_1,T;Q)}, \Delta w_{r_{2_{i_2}}}^{(s_2,T;Q)}, \Delta w_{r_{1_{i_2}}}^{(s_1,T;Q)}, \Delta w_{r_{2_{i_1}}}^{(s_2,T;Q)}$, $\Delta w_{r_{1_{i_1}}}^{(s_2,T;Q)}, \Delta w_{r_{1_{i_2}}}^{(s_2,T;Q)}, \Delta w_{r_{2_{i_1}}}^{(s_1,T;Q)}$ and, $\Delta w_{r_{2_{i_2}}}^{(s_1,T;Q)}$, are independent to each other.

Now, from statement (i) of Lemma I.1, for any $s_1 \in S_{o_1}, p_{o_1}^{(s_1,T)} = 1$ and $\forall x^{(j)} = o_1$ for some $j \in [n], G_j^{(s_1,T)} \geq \frac{1}{2}$. Furthermore, from statement (iii) of Lemma I.1, $\sigma_{o_1}^{(s_1,T)} = \Omega(mlC_2\sqrt{\frac{\log l}{C_p}})$.

Therefore, for any $(x,y) \sim \mathcal{D}$ such that $\exists j \in [n]$ with $x^{(j)} = o_1$,

$\sum_{s_1 \in S_{o_1}} f_{s_1}^{(T)}(x) = \Omega(\gamma_{o_1} mlC_2\sqrt{\frac{\log l}{C_p}})$.

On the other hand, from statement (i) of Lemma I.1, for any $s_3 \in S_{-o_1}, p_{o_1}^{(s_3,T)} = 0$.

Therefore, for any $(x,y) \sim \mathcal{D}$ such that $\exists j \in [n]$ with $x^{(j)} = o_1$,

$\sum_{s \in S_+} f_s^{(T)}(x) = \Omega(\gamma_{o_1} kmlC_2\sqrt{\frac{\log l}{C_p}})$.

Again, from statement (iv) of Lemma I.1, for any $q \in \mathcal{P}\backslash\{o_1, o_2\}, \forall s \in [k], \sigma_q^{(s,T)} = O(mC_2)$, and $\forall s \in S_-, \sigma_{o_1}^{(s,T)} = O(mC_2)$.

Therefore, for any $(x,y) \sim \mathcal{D}$ such that $\exists j \in [n]$ with $x^{(j)} = o_1$,

$\sum_{s \in S_+} f_s^{(T)}(x) - \sum_{s \in S_-} f_s^{(T)} = \Omega(\gamma_{o_1} kmlC_2\sqrt{\frac{\log l}{C_p}}) - O(klmC_2)$, which implies for any $(x,y) \sim \mathcal{D}$ such that $\exists j \in [n]$ with $x^{(j)} = o_1, y f^{(T)}(x) > 0$.

Therefore, to ensure that for any $(x,y) \sim \mathcal{D}$ such that $\exists j \in [n]$ with $x^{(j)} = o_1, y f_Q^{(T)}(x) > 0$, we need $\langle w_r^{(s_1,T)} - w_r^{(s_1,T;Q)}, o_1 \rangle \leq O(lC_2\sqrt{\frac{\log l}{C_p}})$, for all $r \in [m]$ of all $s_1 \in S_{o_1}$ that satisfy $\langle w_r^{(s_1,0)}, o_1 \rangle \geq 0$.

Now, for an $r \in [m]$ of an $s_1 \in S_{o_1}$,

$$\mathbb{P}\left[\left|\langle w_r^{(s_1,T)} - w_r^{(s_1,T;Q)}, o_1\rangle\right| \geq \Omega(lC_2\sqrt{\frac{\log l}{C_p}})\right] \leq \exp\left(-\frac{\Omega(l^2 C_2^2 \log l/C_p)}{d\Delta_r^{(s_1)^2}}\right).$$

Therefore, for all $r \in [m]$ of all $s_1 \in S_{o_1}$, we need $\Delta_r^{(s_1)} \leq O(lC_2\sqrt{\frac{\log l}{C_p d \log(\gamma_{o_1} kmd^2)}})$ to ensure that, for all $r \in [m]$ of all $s_1 \in S_{o_1}$, we have,

$$\mathbb{P}\left[\langle w_r^{(s_1,T)} - w_r^{(s_1,T;Q)}, o_1\rangle \leq O(lC_2\sqrt{\frac{\log l}{C_p}})\right] \geq 1 - \frac{1}{d^2}.$$

Similarly, for all $r \in [m]$ of all $s_2 \in S_{o_2}$, we need $\Delta_r^{(s_2)} \leq O(lC_2\sqrt{\frac{\log l}{C_p d \log(\gamma_{o_2} kmd^2)}})$ to ensure that, for all $r \in [m]$ of all $s_2 \in S_{o_2}$, we have

$$\mathbb{P}\left[\langle w_r^{(s_2,T)} - w_r^{(s_2,T;Q)}, o_2\rangle \leq O(lC_2\sqrt{\frac{\log l}{C_p}})\right] \geq 1 - \frac{1}{d^2},$$

for all $r \in [m]$ of all $s_3 \in S_{-o_1}$, we need $\Delta_r^{(s_3)} \leq O(\frac{(1-\alpha)}{\alpha}l^2 C_2\sqrt{\frac{\log l}{C_p d \log(\gamma_{-o_1} kmd^2)}})$ to ensure that, for all $r \in [m]$ of all $s_3 \in S_{-o_1}$, we have

$$\mathbb{P}\left[\langle w_r^{(s_3,T)} - w_r^{(s_3,T;Q)}, -o_1\rangle \leq O(\frac{(1-\alpha)}{\alpha}l^2 C_2\sqrt{\frac{\log l}{C_p}})\right] \geq 1 - \frac{1}{d^2}, \text{ and}$$

for all $r \in [m]$ of all $s_4 \in S_{-o_2}$, we need $\Delta_r^{(s_4)} \leq O(\frac{(1-\alpha)}{\alpha}l^2 C_2\sqrt{\frac{\log l}{C_p d \log(\gamma_{-o_2} kmd^2)}})$ to ensure that, for all $r \in [m]$ of all $s_4 \in S_{-o_2}$, we have

$$\mathbb{P}\left[\langle w_r^{(s_4,T)} - w_r^{(s_4,T;Q)}, -o_2\rangle \leq O(\frac{(1-\alpha)}{\alpha}l^2 C_2\sqrt{\frac{\log l}{C_p}})\right] \geq 1 - \frac{1}{d^2}.$$

Now, for all $r \in [m]$ of all $s \in S_-$, if $\Delta_r^{(s)} = \max\left\{\Delta_r^{(s_1)}, \Delta_r^{(s_3)}\right\}$, we have $\forall (x,y) \sim \mathcal{D}$ such that $\exists j \in [n]$ with $x^{(j)} = \pm o_1$,

$$\mathbb{P}\left[-\sum_{s \in S_-} f_{Q_s}^{(T)}(x) = O(\sqrt{kml}C_2\sqrt{\frac{\log l}{C_p}})\right] \geq 1 - \frac{1}{d^2}.$$

Here, $f_{Q_s}^{(T)}(x)$ is the quantized output for the expert $s$.

Similarly, for all $r \in [m]$ of all $s \in S_1$, if $\Delta_r^{(s)} = \max\left\{\Delta_r^{(s_2)}, \Delta_r^{(s_4)}\right\}$, we have $\forall (x,y) \sim \mathcal{D}$ such that $\exists j \in [n]$ with $x^{(j)} = \pm o_2$,

$$\mathbb{P}\left[\sum_{s \in S_+} f_{Q_s}^{(T)}(x) = O(\sqrt{kml}C_2\sqrt{\frac{\log l}{C_p}})\right] \geq 1 - \frac{1}{d^2}.$$

Therefore, for all $s_1 \in S_{o_1}, s_2 \in S_{o_2}, s_3 \in S_{-o_1}, s_4 \in S_{-o_2}$, for all $r \in [m]$, we need $\Delta_r^{(s_1)} = O(lC_2\sqrt{\frac{\log l}{C_p d \log(\gamma_{o_1} kmd^2)}}), \Delta_r^{(s_2)} = O(lC_2\sqrt{\frac{\log l}{C_p d \log(\gamma_{o_2} kmd^2)}})$, and

$$\Delta_r^{(s_3)} = O(\frac{(1-\alpha)}{\alpha}l^2 C_2\sqrt{\frac{\log l}{C_p d \log(\gamma_{-o_1} kmd^2)}}),$$

$$\Delta_r^{(s_4)} = O(\frac{(1-\alpha)}{\alpha}l^2 C_2\sqrt{\frac{\log l}{C_p d \log(\gamma_{-o_2} kmd^2)}}).$$

Now, as for all $s \in [k]$, $\max_{r \in [m]} \text{Var}(w_r^{(s,T)}) = \Theta(1)$. On the other hand, for any $s \in [k]$ and any $r \in [m]$, using the Von-Szokefalvi-Nagy inequality, $\text{Var}(w_r^{(s,T)}) \geq \dfrac{\beta_r^{(s,T)^2}}{2d}$. Therefore, for all $s \in [k]$, $\max_{r \in [m]} \beta_r^{(s,T)} = \Theta(\sqrt{d})$.

Let us denote the bit-width of the expert $s_1 \in S_{o_1}, s_2 \in S_{o_2}, s_3 \in S_{-o_1}$, and $s_4 \in S_{-o_2}$ by $b_{s_1}, b_{s_2}, b_{s_3}$, and $b_{s_4}$, respectively.

Therefore, we need
$$b_{s_1} = \log_2\left(1 + \frac{\max_{r \in [m]} \beta_r^{(s_1,T)}}{\min_{r \in [m]} \Delta_r^{(s_1)}}\right) \geq \log_2\left(1 + \Omega\left(\frac{d}{lC_2}\sqrt{\frac{C_p \log(\gamma_{o_1}kmd^2)}{\log l}}\right)\right).$$

Similarly, we need $b_{s_2} \geq \log_2\left(1 + \Omega\left(\dfrac{d}{lC_2}\sqrt{\dfrac{C_p \log(\gamma_{o_2}kmd^2)}{\log l}}\right)\right)$,
$$b_{s_3} \geq \log_2\left(1 + \Omega\left(\frac{\alpha d}{(1-\alpha)l^2 C_2}\sqrt{\frac{C_p \log(\gamma_{-o_1}kmd^2)}{\log l}}\right)\right),$$
and $b_{s_4} \geq \log_2\left(1 + \Omega\left(\dfrac{\alpha d}{(1-\alpha)l^2 C_2}\sqrt{\dfrac{C_p \log(\gamma_{-o_2}kmd^2)}{\log l}}\right)\right).$

Now, from statement (ii) of Lemma I.1, by selecting $\kappa \geq \gamma$, we can ensure that $\forall s_1 \in S_{o_1}$, and $\forall s_2 \in S_{o_2}, b_{s_1}, b_{s_2} = b_h$.

As $\gamma_{o_1}, \gamma_{o_2}, \gamma_{-o_1}, \gamma_{-o_2}$ are $\Omega(1)$, we need

$b_h \geq \log_2(1 + \Omega(d\sqrt{C_p \log(kmd^2)/l^2 C_2^2 \log l}))$, and

$b_l \geq \log_2(1 + \dfrac{\alpha}{1-\alpha}\Omega(d\sqrt{C_p \log(kmd^2)/l^2 C_2^2 \log l}))$, to ensure that,

$\mathbb{P}[\forall (x,y) \sim \mathcal{D} : yf_Q^{(T)}(x) > 0] = 1$

$\square$

