# OpenReview forum: "Efficient Quantization of Mixture-of-Experts with Theoretical Generalization Guarantees"
_ICLR.cc/2026/Conference — ICLR 2026 Poster_

### Official Review · Reviewer_CgYX · 2025-10-27

**Soundness:** 3
**Presentation:** 3
**Contribution:** 3
**Rating:** 6
**Confidence:** 2

**Summary:**

This paper proposes a novel quantization method for Mixture-of-Experts (MoE) models that dynamically selects the precision for each expert based on changes in the $l_2$ norm of its weights. Theoretical analysis is provided for a toy MoE architecture, and numerical experiments demonstrate that the proposed approach outperforms heuristic methods in low-bit quantization scenarios.

**Strengths:**

1. Theoretical results are provided, and the method design is well aligned with the theoretical justification.
2. Numerical results demonstrate that the proposed method achieves performance gains over baseline approaches, delivering comparable or even superior performance to heuristic methods and techniques that rely on calibration sets.

**Weaknesses:**

1. The theoretical analysis is based on a highly simplified MoE architecture focused on binary classification tasks. The gap between the theory and empirical results is not sufficiently addressed. In particular:
   (1) Can the insights from Theorem 4.4 be used to *adaptively* determine quantization precision—rather than relying on predefined bit levels?
   (2) Do activations in practical MoE models align with the conclusions of Theorem 4.3? If not, what are the key discrepancies, and how do they impact the method’s effectiveness?

2. Several important experimental details are missing:
   (1) In Figure 2, the “activation weights” baseline—which specific method  does it correspond to?
   (2) How are the hyperparameters (e.g., $\zeta$) selected? Is their choice task- or model-dependent, and how sensitive is the final performance to variations in $\zeta$?

**Questions:**

See weakness

---

> ### Author Response · Authors · 2025-11-25
>
> ### (Response to Weakness 1) **Our analytical setup is the state-of-the-art in theoretical generalization analysis of MoE.**
>
> The theoretical optimization and generalization analysis of a network with more than two layers is extremely challenging (mainly due to the extreme nonconvexity of the architecture and the intractability of the learned feature; even the two-layer setting is highly nonconvex). Therefore, the two-layer setting remains the state-of-the-art in theoretical optimization and generalization analysis of NN for the feature-learning framework [1,2,3], including the prior works on MoE [4,5,6].
>
> Our theoretical analysis can be directly extended to multiclass classification by analyzing either the cross-entropy loss or the multiclass Hinge loss. However, as the expression of the output logit for each class is the same as in equation (5), it would lead us to the same conclusion and insights as for the binary classification presented in the paper.
>
> [1] Zeyuan Allen-Zhu and Yuanzhi Li. Feature purification: How adversarial training performs robust deep learning. In 2021 IEEE 62nd Annual Symposium on Foundations of Computer Science (FOCS), pages 977–988. IEEE, 2022.
>
> [2] Zeyuan Allen-Zhu and Yuanzhi Li. Towards understanding ensemble, knowledge distillation and self-distillation in deep learning. In The Eleventh International Conference on Learning Representations, 2023.
>
> [3] Dake Bu, Wei Huang, Taiji Suzuki, Ji Cheng, Qingfu Zhang, Zhiqiang Xu, and Hau-San Wong. Provably neural active learning succeeds via prioritizing perplexing samples. In Proceedings of the 41st International Conference on Machine Learning, pages 4642–4695, 2024.
>
> [4] Chen, Z., Deng, Y., Wu, Y., Gu, Q., & Li, Y. (2022). Towards understanding mixture of experts in deep learning. arXiv preprint arXiv:2208.02813.
>
> [5] Chowdhury, M. N. R., Zhang, S., Wang, M., Liu, S., & Chen, P. Y. (2023, July). Patch-level routing in mixture-of-experts is provably sample-efficient for convolutional neural networks. In International Conference on Machine Learning (pp. 6074-6114). PMLR.
>
> [6] Chowdhury, M. N. R., Wang, M., El Maghraoui, K., Wang, N., Chen, P. Y., & Carothers, C. (2024, July). A provably effective method for pruning experts in fine-tuned sparse mixture-of-experts. In Proceedings of the 41st International Conference on Machine Learning (pp. 8815-8847).

---

> > ### Author Response · Authors · 2025-11-25
> >
> > ### (Continued Response to Weakness 1) **Our theoretical insights are verifiable in practical MoE models**
> >
> > For accurate verification of our theoretical insights, we first need to identify the task-relevant tokens of different experts, which is hard to determine in practical MoE models. This is because, for some experts, many tokens can be task-relevant, but each of them may appear less frequently in data. On the contrary, for some experts, only a few tokens can be relevant, but each of them may appear very frequently in data. We consider the tokens with high *gating values* as the task-relevant tokens of each expert.
> >
> > **Larger router norm experts exhibit higher activation.** We verify our claim that larger router norm experts exhibit higher activation in practical MoE models. To verify that claim, we plot the average activations of tokens with top gating values (sampled from the WikiText2 dataset) for each expert in the first five layers of Mixtral8x7B. The results are provided in this anonymous link (https://storage.googleapis.com/anonymous1/activation_visualization.pdf). As we can see, the largest router norm expert (in some cases, the largest two) always exhibits significantly large average activation compared to other experts, except for the second layer. In this case, the lowest one has unusually high activation. However, from our maximum-intra neuron variance, i.e., $\mathrm{MaxVar}$ visualization given in this anonymous link: (https://storage.googleapis.com/anonymous1/maxvar_visualization.pdf ), we can see that this expert has an unusually large $\mathrm{MaxVar}$ value than other experts. Therefore, this expert will be placed on higher bit regardless of its position in the router norm order according to our method.
> >
> > **Verifying the conclusion of Lemma 4.3 and addressing the gap.** We approximated the activation ratio (the quantity of interest for verifying the conclusion in Lemma 4.3) between the largest router norm expert and the largest activation experts among the remaining ones in the activation plots provided above for layer 2, layer 3, and layer 4. They are 1.06, 1.13, and 1.04, respectively. However, our approximation of $\alpha$ (also estimated from the plots; 0.122, 0.12, and 0.117, respectively) indicates the activation ratio should be around 3.09, 3.17, and 3.27, respectively. We believe the gap is arising from our input data distribution assumption (see our data model in section 4.2), where we assume that the tokens are noise-free (i.e., the orthogonality assumption). On the other hand, in practice, the input tokens in an MoE layer are often noisy. However, it is worthwhile to mention that the orthogonality assumption has been widely adopted in theoretical generalization analysis of NN [7, 8, 9, 10], and indeed necessary to facilitate the analysis.
> >
> > **Token visualization.** Finally, we visualize the top *gating value* tokens (sampled from the WikiText2 dataset) through their corresponding model input embeddings for the first MoE block (closest to the model input embeddings) of Mixtral8x7B. The visualization of the first 100 tokens with top gating values (highlighted in yellow), along with their adjacent tokens, is provided in this anonymous link (https://storage.googleapis.com/anonymous1/token_visualization.pdf).
> >
> > As we can see, the lowest router norm expert (Expert-7) activates on subwords of unusual names/nouns (e.g.,  batrachich**ni**,  prolac**erti**form, Chiz**ad**, Oxaz**iri**dine, Moroc**co**, Amalgam**ation**, Straggl**ers**,  hect**ares**, Tenoch**tit**lán,  Embolom**eres**). Each of them is rare in data, but can be critical in the context. This verifies our “critical but important” hypothesis. On the other hand, the highest router norm expert (Expert-6) activates on many common full words of the English language, such as pronouns (e.g., **that**), prepositions (e.g., **to**, **for**, **in**, **at**, **of**, **on**, **with**, **from**), conjunctions (e.g., **and**), and present participle verbs (e.g., depict**ing**). This verifies our claim that the larger router norm experts learn “more frequent” tokens.
> >
> > [7] Alon Brutzkus and Amir Globerson. An optimization and generalization analysis for max-pooling networks. In Uncertainty in Artificial Intelligence, pp. 1650–1660. PMLR, 2021.
> >
> > [8] Zhenmei Shi, Junyi Wei, and Yingyu Liang. A theoretical analysis on feature learning in neural networks: Emergence from inputs and advantage over fixed features. In International Conference on Learning Representations, 2022.
> >
> > [9] Zeyuan Allen-Zhu and Yuanzhi Li. Feature purification: How adversarial training performs robust deep learning. In 2021 IEEE 62nd Annual Symposium on Foundations of Computer Science (FOCS), pp. 977–988. IEEE, 2022.
> >
> > [10] Zixiang Chen, Yihe Deng, Yue Wu, Quanquan Gu, and Yuanzhi Li. Towards understanding the mixture-of-experts layer in deep learning. In Advances in Neural Information Processing Systems, pp. 23049–23062, 2022b.

---

> > > ### Author Response · Authors · 2025-11-25
> > >
> > > ### (Continued Response to Weakness 1) **Our theoretical insights from Theorem 4.4 can be used to adaptively determine the bit level of lower-bit experts, given the bit-width precision of higher-bit experts.**
> > >
> > > To adaptively determine the bit-width precisions of both higher-bit experts ($b_h$) and lower-bit experts ($b_l$) from equation 11 and 12 (in Theorem 4.4), respectively, we need to determine the hidden constant inside these equations. However, the hidden constant depends on the exact identification of task-irrelevant tokens of each expert and their initial alignment along these tokens at the beginning of training. Therefore, the adaptive determination of $b_h$ and $b_l$ is not feasible in practical MoE models.
> > >
> > > However, given the bit-width precision of higher-bit experts ($b_h$), we can determine the bit-width precision of lower-bit experts ($b_l$) using our conclusion of Theorem 4.4. Specifically, as stated in the discussion after Theorem 4.4, the lower-bit precision ($b_l$) is at least $\log_2((1-\alpha)/\alpha)$ bits lower than $b_h$. From our approximation of $\alpha$ (approximated from the activation-ratio estimated from the activation plots) of layer 2, layer 3, and layer 4, we can infer that the lower-bit expert (i.e., the highest router norm expert) bit-width precision is 1.66, 1.71, and 1.62 bits lower than the corresponding $b_h$ for layer 2, layer 3 and layer 4, respectively.

---

> ### Author Response · Authors · 2025-11-25
>
> ### (Response to Weakness 2) **We select the hyperparameter $\zeta$ in reference to the variance of uniform distribution.**
>
> **Selection of the hyperparameter $\zeta$.** As we introduce the hyperparameter $\zeta$ for the first time in the paper in section 3.3, we provide a footnote (on page 4) about how we select the value in our experiments and also provide the reasoning behind the selection. Specifically, we select $\zeta=3$,  since the variance of any bounded distribution is at most three times that of a uniform distribution with the same range. We elaborate on the point in more detail as follows:
>
> Given a fixed weight range, we can infer from equation 2 in the paper that a more non-uniform weight distribution will produce higher quantization noise. Now, for any bounded distribution, the maximum variance is at most *three* times the variance of the uniform distribution with the same range [11]. Therefore, for any two experts, if one of them has more than *three* times the maximum intra-neuron variance ($\mathrm{MaxVar}$) than the other, the former expert should inject significantly high quantization noise into the model. Therefore, we select $\zeta=3$ universally across models and tasks.
>
> Our selection of $\zeta$ alters the initial router norm based order by a very small amount (only 11 out of 256 experts are reordered in Mixtral 8x7B, and only 28 out of 768 experts are reordered in Switch Transformer for our selection of $\zeta$). Indeed, our selection of $\zeta$ only reorders the experts that have unusually large $\mathrm{MaxVar}$ value in an MoE layer; see the $\mathrm{MaxVar}$ visualization of all the experts of Mixtral8x7B model in this anonymous link (https://storage.googleapis.com/anonymous1/maxvar_visualization.pdf).
>
> We conduct a sweep of $\zeta$ on Mixtral 8x7B on the eight benchmark downstream tasks of Table 1 of the paper and report the average accuracy in the following table:
>
> | Avg. bits/expert  | $\zeta=1.0$ | $\zeta=2.0$ | $\zeta=2.5$ | $\zeta=3.0$ | $\zeta=4.0$ | $\zeta=5.0$ |
> |--------------------|---------------|---------------|----------------|---------------|---------------|---------------|
> | 2.0                      | 60.44           | 61.56           | 62.28            | **62.56**          | 61.32           | 61.74           |
>
> As we can see, the performance picks around $\zeta=3.0$, which justifies our selection.
>
> **Activation weights method.** We briefly describe the activation weights baseline provided in Figure 2 in the paragraph right next to the figure (lines 401 to 404). The method is used for expert-wise mixed-precision quantization of MoE in [12], [13]. The method computes the average *gating values* over all the tokens received by an expert in a calibration dataset (termed as *activation weights* in [12] and [13]) and orders the experts in descending order of the value to place the higher *activation weights* in higher bit.
>
> [11] Popoviciu, T. (1935). "Sur les équations algébriques ayant toutes leurs racines réelles". Mathematica (Cluj). 9: 129–145.
>
> [12] Pingzhi Li, Xiaolong Jin, Yu Cheng, and Tianlong Chen. Examining post-training quantization for mixture-of-experts: A benchmark. arXiv preprint arXiv:2406.08155, 2024.
>
> [13] Wei Huang, Yue Liao, Jianhui Liu, Ruifei He, Haoru Tan, Shiming Zhang, Hongsheng Li, Si Liu, and XIAOJUAN QI. Mixture compressor for mixture-of-experts LLMs gains more. In The Thirteenth International Conference on Learning Representations, 2025a.

---

> > ### Author Response · Authors · 2025-11-27
> > **A Gentle Reminder for Reviewer CgYX's Feedback**
> >
> > Dear Reviewer CgYX,
> >
> > We sincerely appreciate your detailed review and constructive comments. Your feedback has helped us refine and strengthen our submission.
> >
> > We are writing here to gently remind you that the discussion window will close soon, at 11:59 pm AoE on December 2. We want to make sure that there is sufficient time to address any additional questions or concerns you might have before reviewer responses close. After the deadline, reviewers can no longer comment, and we will be unable to respond after 11:59 pm AoE on December 3.
> >
> > If there are any remaining points you would like us to clarify, we would be glad to provide further explanation while the discussion phase remains open. We will be happy to respond to any follow-up you may have.
> >
> > If our rebuttal has satisfactorily addressed the issues raised in your review, we would be grateful if you might consider whether increasing your score would better represent your updated view.
> >
> > Thank you again for your time and helpful insights.
> >
> > Sincerely,
> >
> > Authors

---

### Official Review · Reviewer_LANt · 2025-11-01

**Soundness:** 3
**Presentation:** 3
**Contribution:** 3
**Rating:** 6
**Confidence:** 2

**Summary:**

This paper presents an expert-wise mixed-precision quantization method for MoE models. It allocates higher bit-widths to experts that are more sensitive to quantization, identified through the router’s L2 norm change and neuron variance. The paper provides theoretical support showing that experts learning rare but important features require higher precision. Experiments show that the method achieves well performance than existing quantization methods with lower average bit-width and minimal overhead.

**Strengths:**

1.	The paper proposes a novel and efficient expert-wise mixed-precision quantization strategy based on router norm change.
2.	The paper provides theoretical analysis explaining why experts learning rare features are more sensitive to quantization errors.
3.	Achieves well performance and inference efficiency under the same average bit budget.

**Weaknesses:**

1.	When only pretrained router norms are used without fine-tuning, if the pretraining corpus and downstream data distributions differ, can the norm-based ranking still reliably identify “rare but important” experts?
2.	Can the claim that experts learning rare tokens exhibit weaker activations and smaller router norm changes be directly verified through visualization or statistical analysis on real LLM corpora?
3.	Would lightly retraining the low-bit experts further reduce the average bit-width or improve model robustness?

**Questions:**

see the questions

---

> ### Author Response · Authors · 2025-11-24
>
> ### (Response to Weakness 1) **When the downstream data distribution differs from the pretraining corpus, the pretrained router norms without finetuning can reliably identify the appropriate lower-bit and higher-bit experts.**
>
> We consider two cases of the downstream data distribution differing from the pretraining corpus and explain why the pretrained router norms without finetuning can reliably identify the appropriate lower-bit and higher-bit experts.
>
> **Case-1:** The downstream task-relevant features are different than the relevant features of the pretraining corpus and can be viewed as noisy versions of the pretraining relevant features.
>
> **Case-2:** The *less frequent* and *more frequent* features are swapped in the downstream task from the pretraining corpus
>
> Note that the activation pattern for different tokens remains the same across experts, as we are not finetuning the model. Therefore, the lower router norm experts will still exhibit weaker activation than the higher router norm experts for the downstream task.
>
> Now, for **Case-1**, as we justify in our paper, the lower router norm experts are more sensitive to noise due to their weaker activation; these experts should still be assigned to higher bits. Therefore, the pretrained router norms can reliably identify the appropriate lower-bit and higher-bit experts for Case-1.
>
> Now, for **Case-2**, the alteration of *less frequent* and *more frequent* features will not change their activation pattern for their corresponding experts, as we are not finetuning the model on the new downstream dataset. Therefore, relative sensitivity to quantization noise remains the same across different experts for the downstream data. Therefore, the pretrained router norms can reliably identify the appropriate lower-bit and higher-bit experts for Case-2 also.
>
> Our experimental results in Table 1 indeed support the above reasoning, as we can see that the pretrained router norm based method outperforms other competitive baselines in most of the downstream tasks.

---

> > ### Author Response · Authors · 2025-11-24
> >
> > ### (Response to Weakness 2) **Our theoretical insights are verifiable in practical MoE models.**
> >
> > We first want to draw attention to two important points:
> >
> > (i) Our classification of *less frequent* and *more frequent* tokens is measured over the important tokens (i.e., the *task-relevant* tokens in section 4) learned by different experts, not over all tokens of the input space of an MoE layer. An expert may receive many tokens, but not all of them may be relevant to the task. This is more true for token-choice routing, such as in Switch Transformer or Mixtral 8x7B model, as every token must be routed to an expert. For accurate verification of our theoretical insights, we first need to identify the task-relevant tokens of different experts, which is hard to determine in practical MoE models. This is because, for some experts, many tokens can be task-relevant, but each of them may appear less frequently in data. On the contrary, for some experts, only a few tokens can be relevant, but each of them may appear very frequently in data.
> >
> > (ii) Our reference of input token is on the input space of an MoE layer, not on the input embedding space of the whole model. The input space of an MoE layer can be very different than the model’s embedding space, as the model inputs are processed by many layers before arriving at the MoE layer. Even the inputs of the first MoE block are the outputs of the first multihead attention block, plus the residual stream of the model inputs. Therefore, direct visualization of the input tokens of different experts is not possible.
> >
> > To verify our theoretical insights in practical MoE models, we consider the tokens with high *gating values* as the task-relevant tokens of each expert. We indirectly interpret these tokens (sampled from the WikiText2 dataset) through their corresponding model input embeddings for the first MoE block (closest to the model input embeddings) of Mixtral8x7B. The visualization of the first 100 tokens with top gating values (highlighted in yellow), along with their adjacent tokens, is provided in this anonymous link (https://storage.googleapis.com/anonymous1/token_visualization.pdf).
> >
> > As we can see, the lowest router norm expert (Expert-7) activates on subwords of unusual names/nouns (e.g.,  batrachich**ni**,  prolac**erti**form, Chiz**ad**, Oxaz**iri**dine, Moroc**co**, Amalgam**ation**, Straggl**ers**,  hect**ares**, Tenoch**tit**lán,  Embolom**eres**). Each of them is rare in data, but can be critical in the context. This verifies our *critical but infrequent* hypothesis. On the other hand, the highest router norm expert (Expert-6) activates on many common full words of the English language, such as pronouns (e.g., **that**), prepositions (e.g., **to**, **for**, **in**, **at**, **of**, **on**, **with**, **from**), conjunctions (e.g., **and**), and present participle verbs (e.g., depict**ing**). This verifies our claim that the larger router norm experts learn *more frequent* tokens.
> >
> > Finally, we verify our claim that larger router norm experts exhibit higher activation in practical MoE models. To verify that claim, we plot the average activations of tokens with top gating values for each expert in the first five layers of Mixtral8x7B. The results are provided in this anonymous link (https://storage.googleapis.com/anonymous1/activation_visualization.pdf). As we can see, the largest router norm expert (in some cases, the largest two) always exhibits significantly large average activation compared to other experts, except for the second layer. In this case, the lowest one has unusually high activation. However, from our maximum-intra neuron variance, i.e., $\mathrm{MaxVar}$ visualization given in this anonymous link: (https://storage.googleapis.com/anonymous1/maxvar_visualization.pdf), we can see that this expert has an unusually large $\mathrm{MaxVar}$ value than other experts. Therefore, this expert will be placed on higher bit regardless of its position in the router norm order according to our method.

---

> ### Author Response · Authors · 2025-11-24
>
> ### (Response to Weakness 3) **Further quantization-aware retraining of lower router norm experts should reduce the average bit-width requirement of the model.**
>
> It has been established in the literature that quantization-aware training can improve the model's robustness against quantization noise when deployed in low bit-width precision [1], [2], [3]. The method adds noise to the model parameters during the training process so that the model becomes robust to quantization noise after training. However, training large MoE LLMs is computationally expensive. Therefore, only training the most sensitive experts can be a feasible solution. On the other hand, our method indicates that the lower router norm experts are more sensitive to quantization noise. Therefore, light retraining of the lower router norm experts by adding noise to these experts should reduce the average bit-width requirement of the model. We elaborate on the method in more detail as follows.
>
> Suppose an MoE model performs well for the average bit-width of $b_{a_1}$ but loses performance for the average bit-width of $b_{a_2}$, where $b_{a_1}>b_{a_2}$. This is because some of the lower router norm experts are placed from higher-bit to lower-bit to reduce the average bit-width from $b_{a_1}$ to $b_{a_2}$. Therefore, light retraining of these experts by adding noise to their parameters should make them more robust to assign them in lower bit-width precision.
>
> We now provide empirical evidence to support the above claim. Specifically, we add quantization noise during the forward pass of the switch transformer while finetuning only a fraction of lower-bit experts (1/3 fraction of lower-bit experts) with lower-router norm for only one epoch. The results are provided in the following table.
>
> | Average bits/expert   | Rouge-2 score |
> |------------------------|---------------|
> | Full-precision (FP-32) | 19.87         |
> | 2.5                    | 19.50         |
> | 2.25                   | 18.01         |
> | 2.25 (retrained)       | 18.72         |
>
> As we can see, only retraining one-third of the lower-bit experts with lower router norm for only one epoch closes the gap between the performance in 2.5 avg. bits and 2.25 avg. bits by 52.34%.
>
> [1] Naigang Wang, Chi-Chun Charlie Liu, Swagath Venkataramani, Sanchari Sen, Chia-Yu Chen, Kaoutar El Maghraoui, Vijayalakshmi Viji Srinivasan, and Leland Chang. Deep compression of pre-trained transformer models. Advances in Neural Information Processing Systems, 35:14140–14154, 2022.
>
> [2] Baohao Liao, Christian Herold, Shahram Khadivi, and Christof Monz. Apiq: Finetuning of 2-bit quantized large language model. In Proceedings of the 2024 Conference on Empirical Methods in Natural Language Processing, pp. 20996–21020, 2024.
>
> [3] Yuling Gu, Oyvind Tafjord, Bailey Kuehl, Dany Haddad, Jesse Dodge, and Hannaneh Hajishirzi. Olmes: A standard for language model evaluations. arXiv preprint arXiv:2406.08446, 2024.

---

> > ### Author Response · Authors · 2025-11-27
> > **A Gentle Reminder for Reviewer LANt's Feedback**
> >
> > Dear Reviewer LANt,
> >
> > We sincerely appreciate your detailed review and constructive comments. Your feedback has helped us refine and strengthen our submission.
> >
> > We are writing here to gently remind you that the discussion window will close soon, at 11:59 pm AoE on December 2. We want to make sure that there is sufficient time to address any additional questions or concerns you might have before reviewer responses close. After the deadline, reviewers can no longer comment, and we will be unable to respond after 11:59 pm AoE on December 3.
> >
> > If there are any remaining points you would like us to clarify, we would be glad to provide further explanation while the discussion phase remains open. We will be happy to respond to any follow-up you may have.
> >
> > If our rebuttal has satisfactorily addressed the issues raised in your review, we would be grateful if you might consider whether increasing your score would better represent your updated view.
> >
> > Thank you again for your time and helpful insights.
> >
> > Sincerely,
> >
> > Authors

---

### Official Review · Reviewer_8VPJ · 2025-11-01

**Soundness:** 2
**Presentation:** 2
**Contribution:** 2
**Rating:** 4
**Confidence:** 3

**Summary:**

This paper proposes a novel expert-wise mixed-precision quantization strategy for Mixture-of-Experts (MoE) models, aiming to reduce their substantial memory footprint without significant performance degradation. The core contribution is a theoretically-grounded, two-stage heuristic for assigning bit-widths to experts. First, experts are ranked based on the change in their router's L2 norm during training $\Delta^T= || w^T||_2 - || w^0||_2$, the paper's theory suggests that experts with a smaller norm change are more sensitive to quantization and thus require higher precision. Second, this ranking is adjusted by promoting experts with high maximum intra-neuron variance to higher ranks to mitigate quantization noise. The authors provide a theoretical analysis for a simplified two-layer MoE model to justify their primary metric. They validate their approach empirically on large-scale models, including Switch Transformer and Mixtral (8x7B and 8x22B), demonstrating superior performance compared to several baselines, including uniform quantization and prior expert-wise methods. A key advantage highlighted is the negligible computational overhead of this bit-assignment method compared to more expensive SOTA approaches like PMQ.

**Strengths:**

The proposed method in this paper for bit-width assignment is computationally trivial, requiring a simple sort based on router norms. This stands in stark contrast to computationally expensive SOTA methods like PMQ, which require extensive calibration. This is a major practical advantage.

The method demonstrates strong performance on challenging benchmarks with large-scale Mixtral models, often outperforming existing expert-wise and non-expert-wise quantization methods.

**Weaknesses:**

1. For the key experiments on pre-trained models, the method abandons its primary metric (change in norm) for a surrogate (final norm). This switch is poorly justified and severs the link to the paper's own theoretical analysis.

2. The `MaxVar` reordering step is not theoretically motivated and feels like an engineered solution to patch deficiencies in the primary metric. The lack of an ablation study makes it impossible to disentangle its effect, obscuring the true source of the performance gains.

**Questions:**

Roughly same as weakness:

- The use of the final router L2 norm as a surrogate for the change in norm is the most critical unsupported step in the paper. Can you provide any empirical evidence, for instance on a smaller model that can be fine-tuned, that these two metrics produce a similar expert ranking?
- The proposed theory posits that experts with smaller router norm changes learn "less frequent but critical features." Can the authors show what types of tokens or inputs are processed by the high-precision vs. low-precision experts in a real-world task, does this align with your "critical but infrequent" hypothesis?

- Please provide an ablation study that evaluates the performance of your method under three conditions: (i) using only the router norm ordering, (ii) using only the MaxVar ordering, and (iii) the proposed combination. This will help understand where the gains are coming from.

---

> ### Author Response · Authors · 2025-11-24
>
> ### (Response to Weakness 1 and Question 1) **The final router norm provides a good approximation of the change in the router’s norm when the initial routers are randomly initialized with small variance.**
>
> As stated in section 3.3, for the experiments on zero-shot evaluation of pre-trained models, we propose to use the final router norm ($||w_s^{(T)}||$) to approximate the change in the router’s norm ($\Lambda_s^{(T)}:=||w_s^{(T)}||-||w_0^{(T)}||$), as the randomly initialized model is not publicly available for computing the initial router norm ($||w_s^{(0)}||$). The rationale behind the approximation comes from the fact that the initial routers are generally initialized randomly with small variance (e.g., parameters of DeepSeekMoE are initialized randomly with variance 0.000036 [1]). In that case, the initial router norm differences among the routers are too small to alter the change of router norm based order when approximated by final router norm.
>
> Specifically, for any two routers (router 1 and router 2), if router 1’s change in norm is larger than router 2’s change in norm, i.e., $\Lambda_1^{(T)}>\Lambda_2^{(T)}$, then
>
> $\Lambda_1^{(T)}-\Lambda_2^{(T)}>0$
>
> $\Rightarrow (||w_1^{(T)}||-||w_1^{(0)}||) - (||w_2^{(T)}||-||w_2^{(0)}||)>0$
>
> $\Rightarrow (||w_1^{(T)}||-||w_2^{(T)}||) - (||w_1^{(0)}||-||w_2^{(0)}||)>0$
>
> Now, due to the small-variance initialization, $\big| ||w_1^{(0)}||-||w_2^{(0)}|| \big|$ is a very small quantity. Therefore, it is highly likely that $||w_1^{(T)}||-||w_2^{(T)}||>0$, as long as $\Lambda_1^{(T)}-\Lambda_2^{(T)}$ is not too close to zero.
>
> Based on the above intuition, we provide a formal theorem to justify the claim:
>
> **Theorem:** *Let the routers of the initial model be randomly initialized from $\mathcal{N}(0,\sigma^2)$ with $\sigma=O(1/d)$. Then, with high probability, for any two routers $s_1, s_2\in[k]$ such that $\Lambda_{s_1}^{(T)}-\Lambda_{s_2}^{(T)}=\Omega(1/\sqrt{d})$ we have $||w_{s_1}^{(T)}||>||w_{s_2}^{(T)}||$.*
>
> The theorem confirms that, for small-variance initialization (i.e., $\sigma=O(1/d)$), the final norm based order preserves the change in norm based order for any two routers unless they are very close to each other (i.e., $\Lambda_{s_1}^{(T)}-\Lambda_{s_2}^{(T)}=O(1/\sqrt{d})$).
>
> We will include the above discussion, along with the proof of the theorem, in the revised manuscript.
>
> **Empirical validation: rank order** We now provide the additional empirical justification of using the final norm for approximating the change in norm based order as follows:
>
> To evaluate the effectiveness of the final norm based ordering when the initial routers are randomly initialized with small variance, we re-initialized the routers of the pre-trained switch transformer randomly from $\mathcal{N}(0,\sigma^2)$ with $\sigma=0.0005$. We finetune the re-initialized model on the CNN/Daily Mail text summarization task and compare the similarity between the final norm based ranking and the change in norm based ranking in terms of *Spearman’s rank correlation coefficient* ($\rho\in[-1,1]$, $\rho\approx1$ implies high rank correlation), and *Kendall's rank correlation coefficient* ($\tau\in[-1,1]$, $\tau\approx1$ implies high rank correlation). The results are provided below:
>
> | Layer       | Spearman’s $\rho$ | Kendall’s $\tau$ |
> |-------------|--------------|-------------|
> | Encoder-1   | 0.9997       | 0.9950      |
> | Encoder-3   | 0.9994       | 0.9900      |
> | Encoder-5   | 0.9995       | 0.9920      |
> | Encoder-7   | 0.9992       | 0.9871      |
> | Encoder-9   | 0.9989       | 0.9851      |
> | Encoder-11  | 0.9990       | 0.9861      |
> | Decoder-1   | 0.9990       | 0.9871      |
> | Decoder-3   | 0.9997       | 0.9960      |
> | Decoder-5   | 0.9995       | 0.9920      |
> | Decoder-7   | 0.9997       | 0.9950      |
> | Decoder-9   | 0.9998       | 0.9960      |
> | Decoder-11  | 0.9999       | 0.9980      |
>
> The high rank correlation (very close to 1) implies that the rank orders using both methods are very similar.
>
> [1] Dai, D., Deng, C., Zhao, C., Xu, R. X., Gao, H., Chen, D., ... & Liang, W. (2024). Deepseekmoe: Towards ultimate expert specialization in mixture-of-experts language models. arXiv preprint arXiv:2401.06066.

---

> ### Author Response · Authors · 2025-11-24
>
> ### (Continued Response to Weakness 1 and Question 1) **The final router norm provides a good approximation of the change in the router’s norm when the initial routers are randomly initialized with small variance**.
>
> **Empirical validation: quantization results**. We now provide the quantization results for both of the methods below:
>
> | Initial Router              | Method         | Avg. bits/expert | Rouge-2 score |
> |-----------------------------|----------------|-------------------|----------------|
> | Original pre-trained router | Full-precision |   32 (FP)           | 19.87          |
> | Random router                  | Full-precision |   32 (FP)           | 19.46          |
> | Random router | Change in norm              |    2.75               | 18.79          |
> | Random router | Change in norm              |    2.5                 | 18.60          |
> | Random router | Change in norm              |    2.25               | 18.37          |
> | Random router | Final norm                       |    2.75               | 18.81          |
> | Random router | Final norm                       |    2.5                 | 18.59          |
> | Random router | Final norm                       |    2.25               | 18.38          |
>
> The full-precision result of the finetuned model which routers are randomly initialized before finetuning, matches the result of the original finetuned model, i.e., the reinitialized model is well-trained. Now, from the mixed-precision results, as expected, due to the high correlation of the rank order between the final norm and the change in norm based method, the scores of both methods are very similar.

---

> ### Author Response · Authors · 2025-11-24
>
> ### (Response to Weakness 2 and Question 3) **The intuition behind the maximum intra-neuron variance ($\mathrm{MaxVar}$) based reordering is also principled on theory, and the step alters the original router norm based order insignificantly in practical MoE models.**
>
> Our theory indicates the requirement of allocating the experts with significantly larger values of $\mathrm{MaxVar}$ than other experts to higher bit-width precision. As stated in Theorem 4.4, the result of the theorem is based on the assumption that $\forall s\in[k], \mathrm{MaxVar}_{s}^{(T)}=\Theta(1)$, i.e., the values of $\mathrm{MaxVar}$ for all the experts are close to each other. Therefore, the experts with significantly larger $\mathrm{MaxVar}$ values need to be assigned to higher bit. The intuition behind the assumption is hidden in the proof of Theorem 4.4. We explain this as follows:
>
> The large value of $\mathrm{MaxVar}$ arises either from large weight range or from highly non-uniform weight distribution. From equation (2) of the paper, we can infer that both of the cases inject high quantization noise. Therefore, a very large intra-neuron variance of an expert indicates a very high quantization noise. To prevent these experts from having large noise, we propose the reordering step.
>
> Our $\mathrm{MaxVar}$ based reordering step alters the original router norm based order insignificantly. Specifically, in Switch Transformer, only 28 out of 768 experts are reordered, and in Mixtral 8x7B, only 11 out of 256 experts are reordered. We provide the visualization of the $\mathrm{MaxVar}$ values of the experts in different layers of the Mixtral8x7B model in this anonymous link: (https://storage.googleapis.com/anonymous1/maxvar_visualization.pdf) . As we can see, there are very few experts with unusually larger $\mathrm{MaxVar}$ across layers. This justifies our assumption in Theorem 4.4.
>
> We finally provide an ablation study among (i) $\mathrm{MaxVar}$ based ordering, (ii) Router norm based ordering, and (iii) Router norm + $\mathrm{MaxVar}$ based ordering on Mixtral8x7B and report the average accuracy across eight different LLM tasks for the two-bit-level (bit-choices: 2, 3) case, and the three-bit-level (bit-choices: 1, 2, 3) case in the following two tables, respectively:
>
> **Table R-1: Comparison among different methods for the two-bit-level case (bit-choices: 2, 3)**
> | Avg. bits/expert | MaxVar | Router norm | Router norm + MaxVar (Our method) |
> |------------------|--------|-------------|------------------------------------|
> | 2.75             | 69.51  | **69.92**       | 69.50                             |
> | 2.625          | **68.51** | 68.35        |        68.40                      |
> | 2.5              | 66.01  | 67.01       | **67.17**                              |
> | 2.375          | 64.24  | 64.97       | **65.31**                              |
> | 2.25             | 63.88  | 63.84       | **64.26**                             |
> | 2.125          | 60.65  | **61.54**      | 61.43                                  |
>
>
> **Table R-2: Comparison among different methods for the three-bit-level case (bit-choices: 1, 2, 3)**
> | Avg. bits/expert | MaxVar | Router norm | Router norm + MaxVar (Our method) |
> |------------------|--------|-------------|------------------------------------|
> | 2.75             | 69.37  | 54.23       | **70.01**                              |
> | 2.5              | 67.90  | 49.78       | **68.38**                              |
> | 2.25             | 63.97  | 48.12       | **65.79**                              |
> | 2.0              | 60.44  | 44.96       | **62.56**                              |
> | 1.75             | 58.11  | 42.92       | **58.95**                              |
>
> As we can see in Table R-1, for the two-bit-level case, only router-norm-based ordering performs better than only $\mathrm{MaxVar}$-based ordering for most of the average bit points.
>
> However, as we can see in Table R-2, for the three-bit-level case, there is an abrupt drop in performance in the only router-norm-based ordering. This is because some of the unusually large $\mathrm{MaxVar}$ experts are placed in 1 bit, which injects an unbearable amount of quantization noise to the model. Reordering these experts (only 11 out of 256) to higher rank completely removes this issue and significantly outperforms the only $\mathrm{MaxVar}$ based method and other competitive baselines provided in Table 1 of the paper.

---

> ### Author Response · Authors · 2025-11-24
>
> ### (Response to Question 3) **Our theoretical insights are verifiable in practical MoE models.**
>
> We first want to draw attention to two important points:
>
> 1. Our classification of “less frequent” and “more frequent” tokens is measured over the important tokens (i.e., the *task-relevant* tokens in section 4) learned by different experts, not over all tokens of the input space of an MoE layer. An expert may receive many tokens, but not all of them may be relevant to the task. This is more true for token-choice routing, such as in Switch Transformer or Mixtral 8x7B model, as every token must be routed to an expert. For accurate verification of our theoretical insights, we first need to identify the task-relevant tokens of different experts, which is hard to determine in practical MoE models. This is because, for some experts, many tokens can be task-relevant, but each of them may appear less frequently in data. On the contrary, for some experts, only a few tokens can be relevant, but each of them may appear very frequently in data.
>
> 2. Our reference of input token is on the input space of an MoE layer, not on the input embedding space of the whole model. The input space of an MoE layer can be very different than the model’s embedding space, as the model inputs are processed by many layers before arriving at the MoE layer. Even the inputs of the first MoE block are the outputs of the first multihead attention block, plus the residual stream of the model inputs. Therefore, direct interpretation of the input tokens of different experts is not possible.
>
> To verify our theoretical insights in practical MoE models, we consider the tokens with high *gating values* as the task-relevant tokens of each expert. We indirectly interpret these tokens (sampled from the WikiText2 dataset) through their corresponding model input embeddings for the first MoE block (closest to the model input embeddings) of Mixtral8x7B. The visualization of the first 100 tokens with top gating values (highlighted in yellow) along with their adjacent tokens is provided in this anonymous link: (https://storage.googleapis.com/anonymous1/token_visualization.pdf).
>
> As we can see, the lowest router norm expert (Expert-7) activates on subwords of unusual names/nouns (e.g.,  batrachich**ni**,  prolac**erti**form, Chiz**ad**, Oxaz**iri**dine, Moroc**co**, Amalgam**ation**, Straggl**ers**,  hect**ares**, Tenoch**tit**lán,  Embolom**eres**). Each of them is rare in data, but can be critical in the context. This verifies our “critical but infrequent” hypothesis. On the other hand, the highest router norm expert (Expert-6) activates on many common full words of the English language, such as pronouns (e.g., **that**), prepositions (e.g., **to**, **for**, **in**, **at**, **of**, **on**, **with**, **from**), conjunctions (e.g., **and**), and present participle verbs (e.g., depict**ing**). This verifies our claim that the larger router norm experts learn “more frequent” tokens.
>
> Finally, we verify our claim that larger router norm experts exhibit higher activation in practical MoE models. To verify that claim, we plot the average activations of tokens with top gating values for each expert in the first five layers of Mixtral8x7B. The results are provided in this anonymous link (https://storage.googleapis.com/anonymous1/activation_visualization.pdf). As we can see, the largest router norm expert (in some cases, the largest two) always exhibits significantly large average activation compared to other experts, except for the second layer. In this case, the lowest one has unusually high activation. However, from our $\mathrm{MaxVar}$ visualization, we can see that this expert has an unusually large $\mathrm{MaxVar}$ value than other experts. Therefore, this expert will be placed on higher-bit regardless of its position in the router norm order according to our method.

---

> > ### Author Response · Authors · 2025-11-27
> > **A Gentle Reminder for Reviewer 8VPJ's Feedback**
> >
> > Dear Reviewer 8VPJ,
> >
> > We sincerely appreciate your detailed review and constructive comments. Your feedback has helped us refine and strengthen our submission.
> >
> > We are writing here to gently remind you that the discussion window will close soon, at 11:59 pm AoE on December 2. We want to make sure that there is sufficient time to address any additional questions or concerns you might have before reviewer responses close. After the deadline, reviewers can no longer comment, and we will be unable to respond after 11:59 pm AoE on December 3.
> >
> > If there are any remaining points you would like us to clarify, we would be glad to provide further explanation while the discussion phase remains open. We will be happy to respond to any follow-up you may have.
> >
> > If our rebuttal has satisfactorily addressed the issues raised in your review, we would be grateful if you might consider whether increasing your score would better represent your updated view.
> >
> > Thank you again for your time and helpful insights.
> >
> > Sincerely,
> >
> > Authors

---

### Author Response · Authors · 2025-11-27
**Updates on the Revised Manuscript**

We have revised the paper to incorporate all reviewers’ comments and suggestions. The updates we made are listed below:

1. We conduct a thorough ablation study among the two steps of expert ranking proposed in the paper; (i) Only router norm based ranking, (ii) Only $\mathrm{MaxVar}$ based ranking, and (iii) the Router norm + $\mathrm{MaxVar}$ based ranking (**our method**). The results are included in **section 5.3** of the revised manuscript.

2. We provide theoretical and empirical justifications for using the *final router norm* as the surrogate for *change in router norm* for zero-shot evaluation of pretrained models, when the initial version of the model is not publicly available. The empirical results are included in **section 5.4** of the revised manuscript. The theoretical proof is included in **Appendix G** of the revised manuscript.

3. We provide an empirical justification for our selection of $\zeta$. The results are included in **Appendix C** of the revised manuscript

4. We verified our theoretical insights in practical MoE models through the activation and token visualization of the Mixtral 8x7B model. The results are included in **Appendix F** of the revised manuscript.

5. We provided a visualization of the *maximum intra-neuron variance* (i.e., $\mathrm{MaxVar}$) of the Mixtral 8x7B model to justify our assumption in Theorem 4.4. The visualization is included in **Appendix E** of the revised manuscript.

---

### Author Response · Authors · 2025-12-03
**A summary of our work, reviewers’ comments, and our responses**

Dear Area Chair,

Thank you for your time to evaluate our manuscript. To assist you in evaluating our paper, here we provide a brief summary of our work, reviewers’ comments, and our responses.
***
### **A brief summary of our work**

**Background:** Sparse Mixture-of-Experts (MoE) requires huge memory for inference. In previous works, post-training quantization has been explored to address the issue. However, a uniform bit-width for all experts significantly degrades performance for ultra-low-bit (e.g., under 3-bit). On the other hand, the diversity of the experts suggests greater potential for expert-wise mixed-precision (i.e., varying bit-width across experts) in the ultra-low-bit scenario—two recent works explored in this direction. However, their approaches are calibration data-dependent heuristics, require substantial computation for bit-width allocation, and overlook the varying sensitivity of model performance (e.g., model accuracy) to the quantization of different experts.

**Our work:** In this paper, through the lens of feature learning dynamics of MoE, we theoretically investigate why and how we can vary bit-width across experts. Our theoretical analysis reveals that, among the experts with maximum intra-neuron weight variance ($\mathrm{MaxVar}$) within an acceptable range, the experts with lower router norm change during training exhibit weaker activation (as they learn less frequent but critical features), and hence, model performance is more sensitive to the quantization of these experts. Furthermore, the experts which maximum intra-neuron weight variance ($\mathrm{MaxVar}$) is unusually large, inject significant quantization noise to the model. Based on these theoretical insights, we design a two-step expert ranking strategy for allocating experts in higher ranks to higher bit-width. Specifically, we primarily place the experts with lower router norm changes to higher ranks, and then re-order some lower-rank experts to higher ranks if their $\mathrm{MaxVar}$ value is significantly larger ($\zeta$ times larger).

**Our contributions:**

(i) Our proposed expert-wise mixed precision strategy is theoretically-grounded, providing insights about why and how we can vary bit-width across experts.

(ii) Our empirical results demonstrate superior performance over other expert-wise and non-expert-wise mixed-precision baselines.

(iii) Our method reduces the inference computation compared to prior methods, and incurs negligible computational overhead to determine expert bit-widths, while the alternative methods require significant GPU computation.
***
### **Reviewers’ acknowledgements of the contributions:**

The reviewers acknowledged the contributions of our work:

**Novel and theoretically-grounded design:**

(i) Reviewer LANt: “The paper proposes a novel and efficient expert-wise mixed-precision quantization strategy based on router norm change”; “The paper provides theoretical analysis explaining why experts learning rare features are more sensitive to quantization errors.”

(ii) Reviewer CgYX: “Theoretical results are provided, and the method design is well aligned with the theoretical justification.”

**Superior performance over other expert-wise and non-expert-wise methods:**

(i) Reviewer 8VPJ: “The method demonstrates strong performance on challenging benchmarks with large-scale Mixtral models, often outperforming existing expert-wise and non-expert-wise quantization methods.”

(ii) Reviewer LANt: “Achieves well performance and inference efficiency under the same average bit budget.”

(iii) Reviewer CgYX: “Numerical results demonstrate that the proposed method achieves performance gains over baseline approaches, delivering comparable or even superior performance to heuristic methods and techniques that rely on calibration sets.”

**Inference efficiency and negligible computational overhead for expert bit assignment:**

(i) Reviewer 8VPJ: “The proposed method in this paper for bit-width assignment is computationally trivial, requiring a simple sort based on router norms. This stands in stark contrast to computationally expensive SOTA methods like PMQ, which require extensive calibration. This is a major practical advantage.”

(ii) Reviewer LANt: “Achieves well performance and inference efficiency under the same average bit budget.”
***

---

> ### Author Response · Authors · 2025-12-03
> **Continuation of the previous comment**
>
> ***
> ### **Reviewers’ comments and our responses:**
> $\textbf{\textcolor{blue}{Reviewer 8VPJ (Rating: 4)}}$
>
> (i) **Justification of using "final router norm" as the surrogate for "change of router norm" in pretrained MoE models:** We clarified that, the *final router norm* is used to approximate the *change of router norm* based expert ranking, when the initial model is not publicly available, as generally the initial models are initialized randomly with small variance. We provided a complete theoretical and empirical justification of our claim. Our empirical results are included in **section 5.4**, and the theoretical proof is included in **Appendix G** of the revised manuscript.
>
> (ii) **Whether the $\mathrm{MaxVar}$ based reordering is theoretically grounded, and how much gain the two steps provide individually:** We clarify that our $\mathrm{MaxVar}$ based reordering also originated from our theoretical analysis, as our Theorem 4.4 is conditioned on a bound of $\mathrm{MaxVar}$. We provide the $\mathrm{MaxVar}$ visualization of the Mixtral 8x7B model to justify that the condition holds in practical MoE models (included in **Appendix E** of the revised manuscript). In section 3.3 of our original manuscript, we provide the theoretical intuition behind the reordering steps. Finally, we provide a thorough ablation study to characterize the importance and individual gain from each step of expert ranking. We include the results of the study in **section 5.3** of the revised manuscript.
>
> (iii) **Verification of our theoretical insights in practical MoE models:** We verify our theoretical insights in practical MoE models through activation plots and token visualization of the Mixtral 8x7B model. Our results are included in **Appendix F** of the revised manuscript.
>
>
> $\textbf{\textcolor{blue}{Reviewer LANt (Rating: 6)}}$
>
> (i) **For pretrained models without finetuning, whether the pretrained router norm based ranking can reliably identify sensitive experts when the downstream data distribution differs from the pretraining corpus:** We responded affirmatively to the question. Specifically, we clarify that if we use the pre-trained model without finetuning, the activation pattern remains unchanged. Therefore, the pretrained router norm can reliably identify sensitive experts, even if the downstream data differs. The superior or comparable performance across diverse downstream tasks on pretrained Mixtral 8x7B and Mixtral 8x22B justifies the claim.
>
> (ii) **Verification of our theoretical insights in practical MoE models:** We provide a similar response as the reviewer 8VPJ
>
> (iii) **Whether the light retraining of lower-bit experts will improve the performance:** We replied affirmatively and provided a clear justification for the claim through empirical evidence.
>
> $\textbf{\textcolor{blue}{Reviewer CgYX (Rating: 6)}}$
>
> (i) **On the simplified setup for theoretical analysis:** We provide references to show that our setup is state-of-the-art for theoretical training dynamic analysis of MoE. However, our theoretical results are verifiable in practical MoE models
>
> (ii) **Verification of our theoretical results in practical models:** We provide a similar response that we provided to other reviewers. We provide an estimate of the activation ratio (in Lemma 4.3) for the practical model and address the gap between the theoretically analyzed model and the practical model. Finally, we demonstrated how our theoretical results in Theorem 4.4 can be used to adaptively determine the bit-width of lower-bit experts if we know the bit-width of higher-bit experts
>
> (iii) **Justification of the selection of $\zeta$:** We clarify that our selection of $\zeta$ is already provided with a brief reasoning behind the selection in section 3.3 of our original manuscript. However, we provide additional empirical results to further justify our selection. The results are included in **Appendix C** of the revised manuscript
> ***
> Best regards,
>
> Authors

---

### Meta-Review · Area_Chair_exDm · 2026-01-05

**Summary:**

The reviewers’ main concerns can be summarized as follows:

1. Whether using the final router norm as a surrogate for the change in router norm is theoretically and empirically justified when the initial model is unavailable.

2. Whether the maximum intra-neuron variance (MaxVar) adjustment is theoretically motivated or merely an empirical heuristic, and how much it contributes relative to the primary router-norm criterion.

3. Whether the paper sufficiently demonstrates, in practical large MoE models, that experts identified as sensitive indeed correspond to rare-but-critical features, and that the theory aligns with observed activation patterns.

4. Requests for ablation studies, visualizations, and additional results to disentangle the effects of different components of the proposed method.

**Reviewer Concerns:**

Concerns Adequately Addressed:

1. The authors provided both formal theoretical justification (small-variance initialization implies rank preservation with high probability) and strong empirical evidence, including near-perfect Spearman and Kendall rank correlations and matched quantization performance between change-in-norm and final-norm rankings.

2. The rebuttal clarifies that MaxVar arises naturally from the conditions of Theorem 4.4 and is not ad hoc. Moreover, the authors added a clean ablation study separating (i) router-norm-only, (ii) MaxVar-only, and (iii) combined strategies, clearly demonstrating the incremental benefit of each step.

3. The authors added activation visualizations, token-level analyses, and additional experiments on Mixtral models that substantiate the claim that experts with smaller router-norm changes correspond to weaker activations and higher sensitivity to quantization.

Minor or Non-Critical Remaining Issues:

1. The theoretical analysis still relies on a simplified two-layer MoE model, which may not capture all nuances of deep MoE architectures. However, this limitation is standard in theory papers of this kind and is mitigated here by unusually strong empirical validation at scale.

**Reviewer Scores:**

Based on the rebuttal quality and added results, I estimate the reviewers’ updated positions as follows:

1. Reviewer 8VPJ: Original rating: 4
Likely increase to 6. The reviewer’s main objections (surrogate metric and lack of ablation) were directly and convincingly addressed.

2. Reviewer LANt: Original rating: 6
Likely unchanged or increase to 8. Already positive; the revisions further strengthen confidence.

3. Reviewer CgYX: Original rating: 6
Likely unchanged or increase to 8. Theoretical–empirical alignment is now clearer and better supported.

---

### Decision · Program_Chairs · 2026-01-26

Accept (Poster)